# Clickable *C*-Glycosyl Scaffold for the Development of a Dual Fluorescent and [^18^F]fluorinated Cyanine-Containing Probe and Preliminary In Vitro/Vivo Evaluation by Fluorescence Imaging

**DOI:** 10.3390/ph15121490

**Published:** 2022-11-29

**Authors:** Julen Ariztia, Kamal Jouad, Valérie Jouan-Hureaux, Julien Pierson, Charlotte Collet, Bertrand Kuhnast, Katalin Selmeczi, Cédric Boura, Sandrine Lamandé-Langle, Nadia Pellegrini Moïse

**Affiliations:** 1Université de Lorraine, CNRS, L2CM, F-54000 Nancy, France; 2Université de Lorraine, CNRS, CRAN, F-54000 Nancy, France; 3NancycloTEP, Molecular Imaging Platform, CHRU-Nancy, Université de Lorraine, Nancy F-54000, France; 4Université de Lorraine, Inserm, IADI, F-54000 Nancy, France; 5Laboratoire d’Imagerie Biomédicale Multimodale Paris-Saclay, CEA, CNRS, Inserm, Université Paris-Saclay, F-91401 Orsay, France

**Keywords:** *C*-glycosyl compounds, *c*(RGDfK), cyanine-5, fluorine-18, fluorescence, PET, optical imaging, bimodal imaging

## Abstract

Considering the individual characteristics of positron emission tomography (PET) and optical imaging (OI) in terms of sensitivity, spatial resolution, and tissue penetration, the development of dual imaging agents for bimodal PET/OI imaging is a growing field. A current major breakthrough in this field is the design of monomolecular agent displaying both a radioisotope for PET and a fluorescent dye for OI. We took advantage of the multifunctionalities allowed by a clickable *C*-glycosyl scaffold to gather the different elements. We describe, for the first time, the synthesis of a cyanine-based dual PET/OI imaging probe based on a versatile synthetic strategy and its direct radiofluorination via [^18^F]F-C bond formation. The non-radioactive dual imaging probe coupled with two *c*(RGDfK) peptides was evaluated in vitro and in vivo in fluorescence imaging. The binding on α_v_β_3_ integrin (IC_50_ = 16 nM) demonstrated the efficiency of the dimeric structure and PEG linkers in maintaining the affinity. In vivo fluorescence imaging of U-87 MG engrafted nude mice showed a high tumor uptake (40- and 100-fold increase for orthotopic and ectopic brain tumors, respectively, compared to healthy brain). In vitro and in vivo evaluations and resection of the ectopic tumor demonstrated the potential of the conjugate in glioblastoma cancer diagnosis and image-guided surgery.

## 1. Introduction

As a pillar of diagnosis and patient care, molecular imaging is a field of great interest. Early diagnosis could avoid morbidity and medical expenses. Moreover, molecular imaging allows the observation of in vitro or in vivo cellular and molecular processes [1,2]. Specificity of molecular probes for biomarkers or receptors is obviously the corner stone of molecular imaging and is most of the time achieved by vector targeting. Molecular imaging techniques, such as positron emission tomography (PET) [3,4], single-photon emission computed tomography (SPECT), magnetic resonance imaging (MRI), and optical imaging (OI) [5,6], have improved over the years. However, these techniques also display limitations in spatial and temporal resolution or sensitivity. The trend for the last ten years has been the development of bimodal imaging approaches, with the most expanded combinations being PET/MRI and PET/OI. PET/MRI displays high penetrability, but the pitfall of this combination is the low sensitivity of MRI which requires a higher concentration of contrast agent (µM) compared to PET (nM). PET/OI brings together two techniques having a sensitivity in the same range (nM), which is obviously an advantage [7,8]. The spatial resolution of OI (submillimeter) boosts the resolution of PET which is up to the millimeter. Low tissue penetration depth of emitted light photons of OI is a limitation that is fortunately covered by the deep penetrability of PET. Actually, considering the similarity and complementarity of PET and OI, the PET/OI combination couples the non-invasive whole-body diagnosis of PET imaging with the intraoperative imaging-guided surgery or ex vivo histopathology of OI. To achieve this combination, a current major breakthrough is the design of monomolecular dual agents, i.e., single molecules displaying a radioisotope for PET and a fluorescent dye for OI [7,8].

Radiometals, including Gallium-68, Copper-64, and Zirconium-89, are the most encountered radionuclides in the preparation of dual PET/OI imaging agents [7,9]. In this case, the approach to construct the dual agents is based on a step-by-step combination of a fluorescent dye, a vector, and a chelating entity, with the complexation of the radiometal occurring in the last step.[10] Fluorine-18 (^18^F) is also employed in the development of dual PET/OI agents. Indeed, ^18^F draws particular attention considering the metabolic stability of the C–F and B–F bonds and its half-life (t_1/2_ = 109.8 min) being suitable for fast biodistribution vectors, such as peptides [11].

For biomedical applications, fluorophores must have specifications, such as red-shifted absorption and emission wavelengths (Near Infrared biological window, NIR), and high quantum yield. To fulfill these requirements, the cyanine derivatives, BODIPYs and xanthene derivatives, are mostly used [7,8,12]. BODIPYs are a class of fluorophores presenting a fluorine in their structure, however, except for azaBODIPY, these fluorophores do not have absorption and emission in the NIR. Several studies have focused on the development of fluorescent ^18^F radiotracers by taking advantage of the difluoro or disubstituted borane entity [13]. Such radiolabeling implies, in most cases, the formation of a [^18^F]F-B bond which can be performed in a late stage in aqueous reaction conditions, and is compatible with sensitive vectors, including peptides [12]. Regarding cyanines, these organic fluorophores have the advantage of low toxicity and excitation and having emission wavelengths optimal for in vivo imaging (λ_ex_ = 645–695 nm, 735–795 nm, and λ_em_ = 660–710 nm, 675–817 nm in PBS or water for cyanine 5 (Cy5) and cyanine 7 (Cy7), respectively). Numerous [^18^F]F-B labeled cyanines have been described [12], notably via radiofluorination of borylated groups, such as tetraphenylpinacolborane and *N*-alkyl,*N-N*-dimethylammonium methyl trifluoroborate (AMBF_3_) grafted on the terminal nitrogen atom or the central methine group [14,15,16,17,18,19,20,21]. To go further, the formation of a [^18^F]F-C bond is undoubtedly a promising approach, considering the great metabolic stability of this bond compared to [^18^F]F-B, which stability is sometimes debated. The [^18^F]F-C bond is also more advantageous in terms of molar activity, being higher than the one obtained during [^18^F]F-B labeling. However, the radiolabeling usually requires harsh reaction conditions in organic solvent and high temperatures, which are not compatible with sensitive targeting vectors, such as peptides. Few examples of fluorination of cyanine-containing compounds via a [^18^F]F-C bond formation had been described, and none of them reported the direct radiofluorination of a cyanine derivative, except one example describing the introduction of ^18^F via a nucleophilic aromatic substitution [22]. Priem et al. [23] developed a [^18^F]F-C cyanine-based probe via a prosthetic strategy, but this dual dye did not possess a conjugation moiety for vector coupling. With a similar strategy, Schwegmann et al. reported the click ligation of a [^18^F]F-1-azido-2-fluoroethane with a sulfonated Cy5 bearing an alkyne moiety [24], and Zettlitz et al. described a sulfonated cyanine conjugated to a modified antibody fragment and bearing a tetrazine linker, enabling the radiofluorination in the last step of the synthesis [25].

An innovative and versatile synthetic strategy for the conception of an original [^18^F]F-C cyanine-based agent could be divided in two stages. The first step would be the conjugation of the fluorophore and radioisotope on a central scaffold, followed by the late-stage introduction of various vectors on this dual probe, thus allowing versatility (Figure 1). In this work, we propose, for the first time, the synthesis of a [^18^F]F-C cyanine-containing dual PET/OI probe suitable for late-stage vector grafting. RDG derivatives targeting integrins were chosen to establish a proof of concept as they are well-known vectors and we opted for the conjugation with two peptide vectors. This work provides on overview of the synthesis of the dual probe, the conjugation with peptides, the in vitro and in vivo biological evaluation, and the direct radiolabeling of the probe via [^18^F]F-C bond formation. According to our previous works on the development of [^18^F]F-C radiotracers based on carbohydrate scaffolds [26] and the controlled and regioselective functionalization of saccharidic derivatives [27,28,29,30,31,32], a *C*-glycosyl compound was selected as the central platform suitable to bring the different elements together.

## 2. Results and Discussion

### 2.1. Synthetic Strategy: A Clickable Scaffold

The corner stone of the synthetic strategy is a platform obtained by the modifications of a *C*-glycosyl derivative that is conveniently substituted to enable the introduction of the key elements at different stages. The main idea is to use copper-catalyzed alkyne-azide cycloaddition (CuAAC) and apply it for fluorophore introduction and in the bioconjugation step [33,34]. To this end, a clickable scaffold was designed with free or temporarily masked triple bonds which could be activated at an appropriate time. Therefore, a γ-D-ribonolactone was selected as the starting compound since this sugar configuration allows an extended spatial distribution of the different arms, i.e., upper and lower face of the central core. A multi-step synthetic strategy provided access to the non-radioactive dual probe (with a ^19^F) and to the radiolabeling precursor (bearing a leaving group) for ^18^F-radiolabeling.

### 2.2. Scaffold Synthesis

The central core was obtained from a sugar γ-lactone being transformed into a *C*-glycosyl derivative. Among the numerous synthetic methods to prepare *C*-glycosyl derivatives, we have developed and used for several years the Wittig reaction on sugar γ-lactones, which gives an efficient access to functionalized *C*-glycosylidene compounds (commonly called *exo*-glycals), and their subsequent stereoselective double bond reduction (Figure 1) [27,35]. Thus, starting from a commercially available D-ribonolactone **1**, the *C*-glycosyl compound **4** was obtained by a three-step synthetic sequence involving hydroxyls protection, Wittig reaction with a methoxycarbonyl phosphorane (Ph_3_P = CHCO_2_Me), and stereoselective hydrogenation [29,31,32]. Zemplen reaction on compound **4** led to the 7-hydroxy derivative **5**, which is a key intermediate for the introduction of the non-radioactive fluorine atom (^19^F) or for the introduction of the leaving group of the precursor which will undergo a nucleophilic substitution during ^18^F-radiolabeling (Figure 2).

Compound **5** was reacted with diethylaminosulfur trifluoride (DAST) for fluorination and the targeted 7-fluoro derivative **6** was obtained at 85% yield. The strategy based on sequential functionalization required the 7-hydroxyl protection prior to its activation with a leaving group. Thus, a silyl protecting group, which is stable under the envisioned experimental conditions and compatible with 4,5-isopropylidene, was selected. The TBDPS ether **7** was obtained at 95% yield via the reaction of **5** with TBDPSCl and imidazole in DMF. The reduction of the methyl ester on compounds **6** and **7** performed by LiAlH_4_ in THF led, respectively, to **8** and **9** in high yields. The resulting primary hydroxyl of compound **8** was etherified by reaction with propargyl bromide in the presence of sodium hydride (3 eq.) in DMF and led to **10** at 95% yield. The same reaction performed on silyl ether **9** required optimization. The amount of NaH (3.0, 1.2 and 0.9 eq.), the solvent (DMF, acetone or THF), and the reaction duration of deprotonation were screened and the optimal yield of **11** (70%) was obtained in 16 h with 1.2 eq. of NaH and 3.0 eq. of propargyl bromide in THF.

Considering the advantages of CuAAC [33,34], we next planned to introduce two other propargyl groups on positions 4 and 5 (see numbering on Figure 1). These two protected alkynes were used for the introduction of two peptide vectors in a subsequent step. The removal of the 4,5-isopropylidene was easily performed on compound **10** by treatment with a TFA/H_2_O mixture and led to compound **12** in a nearly quantitative yield (84%). Looking at the potential lability of TBDPS in acidic medium, attention was required when the reaction was performed on compound **11**. Indeed, the use of TFA/H_2_O mixture did not permit us to obtain **13** in a sufficient yield since a significant amount of 4,5,7-deprotected compound was formed (observed by TLC). Taking advantage of our experience in this type of selective deprotection, we opted for AcOH in H_2_O at 80 °C that provided compound **13** at 54% yield. It should be noted that the formation of approximatively 10% of 4,5,7-deprotected compound could not be avoided and 15% of compound **11** were still remaining in the crude mixture. This selective deprotection was obviously a crucial point in this synthetic strategy, and compound **13** was obtained in a moderate but sufficient yield to proceed further with the synthesis. After careful purification using a silica gel column chromatography, compounds **12** and **13** were etherified with TIPS-propargyl bromide and NaH in DMF or THF, and the corresponding ethers **14** and **15** were obtained at 94% and 78% yields, respectively.

The cyanine derivative **16** was previously obtained by coupling the commercially available cyanine-5-NHS with the 3-azidopropylamine in DMF (Figure 2). Cyanine **16** was engaged in CuAAC reaction, and cycloadducts **17** and **18** were obtained in excellent yields after purification using a column chromatography. The temporary TIPS and both TBDPS protecting groups were efficiently removed by tetrabutylammonium fluoride, leading to compounds **19** and **20**.

### 2.3. Peptide Functionalization and Coupling

At this stage, two parallel syntheses were carried out for the radiolabeling precursor and the non-radioactive compound coupled with RGD derivatives. *c*(RGDfK) was previously derivatized on the *N-*ε-amine lysine side chain with two different NHS-activated spacers bearing a terminal azido group for CuAAC (Figure 3). A hexyl linker and a PEG_4_ linker were chosen as they enable flexibility, and the PEG_4_ linker was intended to increase the hydrophilic character of the final construct. The resulting azido cyclic peptides **21** and **22** were reacted with compound **19** in a mixture of ACN/H_2_O (Figure 3). For CuAAC with two cyclopeptides, particular attention must be paid to the equivalent numbers of reactants. Indeed, catalytic amounts of copper sulfate and sodium ascorbate were not sufficient and 3.0 and 7.5 eq. were, respectively, used to provide cycloadducts in good yields. Compounds **23** and **24** were purified using a size-exclusion chromatography (Sephadex LH20) and were obtained at 42 and 62% yields, respectively, and successfully characterized by HRMS. Their in vitro and in vivo biological properties were then evaluated. The fluorescent monomeric derivative *c*(RGDfK)-Cy5 **25**, i.e., the compound linked to the cyanine-5 directly on the *N*-ε of the lysine residue, was required for in vitro evaluation and was obtained at 59% yield by coupling *c*(RGDfK) with Cy5-NHS in DMF (Figure 4).

### 2.4. Precursor of Radiolabeling and ^18^F-Radiolabeling

The 7-*O*-activated compound was the key intermediate for nucleophilic radiolabeling with ^18^F. In order to avoid competition with other nucleophilic anions from the reaction mixture, the iodide counter anion of the cyanine indolium was removed and replaced by a triflate group, a non-nucleophilic anion. The anionic metathesis was carried out using an Oasis^®^ MCX cartridge (Waters, Milford, MA, USA) with a solution of silver triflate (0.2 M) (Figure 5). The anion exchange was confirmed using a ^1^H NMR spectroscopy, revealing some variation in the chemical shifts of polymethine chain protons H-27, H-25, and H-29 of cyanine (δ = 6.91, 6.61, and 6.21 ppm for compound **20**, 6.78, 6.29, and 6.24 ppm for compound **26**, see ESI for atom numbering). The ^19^F NMR also showed the signal of trifluoromethyl group (see ESI). Regarding the synthesis of the labeling precursor, we first planned to carry out the preparation of the 7-trifluoromethylsulfonate derivative. Despite careful attention toward the experimental conditions, the recovery of the expected 7-*O*-triflate compound was not possible. We moved on to the 7-*O*-methylsulfonate derivative, which was obtained by reaction with methyl sulfonate anhydride in dichloromethane. The expected mesylate **27** was obtained at 38% yield, which is satisfactory given the tedious and careful column chromatography purification required for cyanine-based compounds.

^18^F-radiobeling of **27** was then investigated using both AllInOne (AIO) and TracerLab FxFN synthesizers with the classical method of radiofluorination using the K[^18^F]F-K_222_ complex (Figure 5). After the optimization of the K_222_/K_2_CO_3_ quantities and ratios, the best conditions were 15 mg/1.3 mg in 8/2 acetonitrile/water (*v*/*v*) for AIO and 12 mg/2 mg in 7/3 acetonitrile/water (*v*/*v*) for Tracerlab FxFN, respectively, with a 10 min reaction in both cases. The identity of **[^18^F]19** was confirmed using an analytical radio-HPLC by comparison with the non-radioactive compound **19**, both having identical retention times (see ESI Appendix A). The ^18^F-radiolabeling of **27** was reproducible using the two different synthesizers, and decay-corrected radiochemical yields (RCY) of 13 and 11% as determined by HPLC analyses were obtained with AIO and TracerLab FxFN, respectively. These results establish the proof of concept of direct ^18^F-radiolabeling of a cyanine-containing precursor via a [^18^F]F-C bond formation.

### 2.5. Photophysical Properties

Photophysical properties (absorption, excitation, emission, and relative fluorescence quantum yield) of compounds **23** and **24** were measured in an aqueous PBS buffer solution (pH = 7.4, *T* = 298 K) and compared to the commercially available Cy5 to detect any potential degradation that could occur during the synthesis and purification steps. The data show that compounds **23** and **24** have very close absorption/emission wavelengths and Stokes shifts compared to the reference [36] (λ_abs_/λ_em_ = 648/662 nm, Δ_s fluo_ = 326 cm^−1^ for **23**, λ_abs_/λ_em_ = 644/658 nm, Δ_s fluo_ = 330 cm^−1^ for **24**). Conjugation of the Cy5 derivative to *c*(RGDfK) has a somewhat stronger effect on the quantum yield of compound **23** (*Φ_fluo_* = 8%) but remains slight for product **24** *(Φ_fluo_* = 14%, Cy5 *Φ_fluo_* = 13%) [36].

### 2.6. In Vitro Biological Evaluation

Integrins are cell surface receptors involved in many physiological and pathological processes [37,38]. The most important member of this receptor family is the α_v_β_3_ integrin, which is involved in blood vessel formation (angiogenesis) and is overexpressed in several cancer types (melanoma, glioma, ovarian, and breast cancers). Therefore, visualizing α_v_β_3_ expression is obviously of great interest and the development of α_v_β_3_ imaging agents is still a concern in the field of molecular imaging. RGD peptide derivatives and, in particular, cyclic *c*(RGDfK) show high α_v_β_3_ affinity in vitro and receptor-specific tumor uptake in vivo [39,40].

The affinity of compounds **23** and **24** for α_v_β_3_ integrin was evaluated in the presence of a coated vitronectine (reference ligand) at different concentrations (1.6 nM–20 µM, solid-phase binding assay), and the IC_50_ values were determined (Figure 2). These data demonstrate a high affinity toward α_v_β_3_ integrin for both compounds (IC_50_ of 10 and 16 nM for **23** and **24**, respectively). The positive control compound *c*(RGDfK) showed an IC_50_ value of 40 nM, while the fluorescent monomeric reference compound *c*(RGDfK)-Cy5 (**25**) showed an IC_50_ value of 8542 nM, which is not surprising considering the steric hindrance induced by the cyanine moiety. The negative control compound **19** did not inhibit the binding of α_v_β_3_ integrin to vitronectin.

Cellular uptake was evaluated using a confocal microscopy on U-87 MG spheroids after 1, 4, and 24 h of exposure with conjugates **23** and **24** and *c*(RGDfK)-Cy5 (**25**) at 1 µM. The fluorescence signal appeared to be stable over time and the distribution of the compounds was homogeneous over all cells and throughout the spheroids (Figure 3A). These three compounds were localized rapidly in the cell cytoplasm due to receptor internalization after ligand binding. As expected from photophysical properties, the fluorescence signal was higher for compound **24** compared to **23**. Compound **23** showed a better cellular uptake probably due to its higher hydrophobicity introduced by hexyl linker. The amount of compounds seemed to decrease slightly at 4 h but stayed relatively stable until 24 h and represented 25.8, 14.7, and 28.6 pmoles/10^6^ cells, respectively, for compounds **23**, **24** and **25**. No significant differences were observed between the different compounds, suggesting that the number of *c*(RGDfK) did not seem to affect cell incorporation (compare compounds **23** and **24** to compound **25** in Figure 3B).

### 2.7. In Vivo Fluorescence Imaging

In order to evaluate the in vivo biodistribution and tumor uptake of compound **24**, which has the best photophysical and solubility properties in comparison to compound **23**, mice bearing ectopic or orthotopic brain tumors were imaged in NIRF using a fluorescent small animal imager. Compound **24** showed a more rapid tumor uptake at 1 h post intravenous administration with a maximum fluorescence signal at 4 h (ectopic tumor, Figure 4A) compared to compound **25** uptake. The fluorescence signal of compound **24** decreased slightly after 6 h, but it was significantly high until 24 h; the elimination clearly seems to occur via urinary tract and liver metabolism (Figure 4B). The healthy brain did not fix either compound **24** or compound **25**. For compound **24**, the orthotopic tumor was largely identifiable and a high tumor uptake was observed for both tumors with ratios of 40 and 100 between the orthotopic and ectopic brain tumors and the healthy brain, respectively, whereas a lesser tumor uptake was observed for compound **25** (Figure 4C).

In order to assess the possibility of using compounds **24** or **25** for extemporaneous tumor cell identification during anatomo-pathology analysis, the ectopic tumors were harvested 24 h after administration and observed using confocal imaging (Figure 5). The whole tumors showed more pronounced staining with compound **24** (Figure 5A) compared to compound **25** (Figure 5B). Moreover, compound **24** showed a clearer staining of all tumor cells at high resolution (Figure 5C) compared to compound **25** (Figure 5D).

Because they allow the early detection of pathologies, participation in patient care, and provision of theranostic tools, molecular imaging and, more specifically, bimodal imaging are research fields in strong expansion. Currently, the main trend is the development of molecular imaging agents targeting the specific biomarkers of a pathology. This contribution aims at designing and synthesizing a monomolecular [^18^F]F-C cyanine-containing dual PET/OI imaging probe. The non-radioactive dual probe was conjugated to two RGD derivatives targeting α_v_β_3_ integrin, and the resulting conjugates were evaluated in vitro and in vivo.

As mentioned in the introduction, the direct radiolabeling of a cyanine-containing compound via a [^18^F]F-C bond formation has not yet been reported, except for one particular example via SNAr [22]. Strategically, if a peptide vector is introduced at an early stage in the synthetic sequence, the entire synthesis has to be repeated for any new vector, which limits the versatility of the approach [23,24,25]. On the other hand, conjugating the vector(s) to a structure already carrying the key elements (i.e., the [^18^F]F-C cyanine-based dual imaging probe) in the last step drastically increases the versatility. Besides, several factors must be taken into consideration during the radiosynthesis, such as reaction conditions (organic solvent, high temperature), automation of the radiosynthesis, and potential reactivity of polyene moiety of cyanine [41,42]. These specifications strongly advocate for a late introduction of the vector in the synthesis scheme as well. In this work, we propose, for the first time, the synthesis of a [^18^F]F-C cyanine-based dual PET/OI imaging probe suitable for a late-stage vector grafting. To face this challenge, we selected a *C*-glycosyl moiety that offers sequential functionalization possibilities thanks to its polyhydroxylated structure. Furthermore, the C–C bond at the anomeric position displays acidic and enzymatic hydrolysis resistance, conferring an improved in vivo stability compared to *O*-glycosides [43,44,45]. The considered synthetic pathway employed orthogonal click reactions [33,34] to achieve linkage of the fluorescent entity in the first step and conjugation of two *c*(RGDfK) vectors in a second step. According to our previous works on [^18^F]F-glycosyl tracers for PET imaging, the introduction of the ^18^F is ensured by a nucleophilic substitution on a mesylate derivative introduced on a primary hydroxyl. The ^18^F-radiolabelling step was successfully performed in standard fluorine activation conditions (K_222_/K_2_CO_3_) and decay-corrected RCY of 13% and 11% were obtained using two different synthesizers (AIO and TracerLab FxFN, respectively). This constitutes the proof of concept of the direct radiolabeling of a cyanine-containing compound via [^18^F]F-C bond formation.

Imaging modalities, such as TEP and OI, require proper detection moieties (radioisotope for PET and fluorescent dye for OI), which must be linked to the targeting ligand. This can induce structural modifications of the vector and affect its binding and, consequently, its affinity for the receptor. Small molecules and peptides are the most impacted by the potential steric hindrance induced by imaging moieties. A way to overcome this limitation is to incorporate linkers long enough to ensure spacing between the ligand and the imaging entities. Another way is to enhance the binding by grafting multiple copies of a ligand on a central core (multivalent effect). Indeed, an increase of receptor binding affinity has previously been observed when more than one RGD peptide is grafted [39,46,47,48]. In this work, we exploited both the use of long linkers and the grafting of several peptide vectors to enhance binding and tumor uptake. Two *c*(RGDfK) were, thus, successfully introduced through the two available positions on the central *C*-glycosidic moiety. This approach appeared fruitful with IC_50_ of 10 and 16 nM for dimers **23** and **24**, respectively. These values are noticeably below the IC_50_ of *c*(RGDfK)-Cy5 **25** (8542 nM) and in the same range of, if not slightly lower than, the one obtained for *c*(RGDfK), demonstrating that sufficiently long linkers and divalency are effective in enhancing ligand–target interactions. The nature of the two linkers, i.e., hexyl for **23** and (PEG)_4_ for **24**, did not have a significant impact on the binding. Nevertheless, it seems that due to its hydrophobicity (hexyl linker), compound **23** was internalized by cells in a greater extent than compound **24**. This lower internalization was largely compensated by the fluorescence properties of compound **24**, enabling the easy imaging of cells by confocal microscopy. Moreover, the fluorescence signal could be followed for up to 24 h for both compounds **23** and **24**. The poor solubility of compound **23** in aqueous solution is a pitfall for its in vivo evaluation, explaining the choice of compound **24** for in vivo fluorescence imaging. The ectopic tumor uptake of compound **24** was maximum at 4 h post intravenous administration; this was in contrary to the tumor uptake of *c*(RGDfK)-Cy5 **25**, which decreased rapidly after its administration. PEG modification of compound **24** allowed a better tumoral distribution with a probable elimination via urinary tract and liver metabolism [49]. Interestingly, the healthy brain tissue did not fix compound **24**, but a high staining of the orthotopic brain tumor was observed after 24 h post administration. All biological evaluations highlighted the efficiency of the dimeric-PEG structure **24** to rapidly accumulate in the orthotopic tumor, with a noticeable uptake and a fluorescence signal 40-fold higher in the tumor compared to the healthy brain.

## 3. Materials and Methods

### 3.1. General Information

The solvents and liquid reagents were purified and dried according to recommended procedures. Cy5-NHS was purchased from CHEMFORASE and Oasis^®^ MCX cartridges from Waters (Milford, MA, USA). Thin layer chromatography (TLC) analyses were performed using standard procedures on Kieselgel 60F254 plates (Merck, Kenilworth, NJ, USA). The compounds were visualized using UV light (254 nm), with ninhydrin and/or a methanolic solution of sulfuric acid, and charred. Flash column chromatography was performed using a Puriflash (Interchim, Montluçon, France). c(RGDfK) was purchased from Bachem (Budendorf, Switzerland) with >95% purity. The purification of RGD-conjugates was achieved using size-exclusion chromatography on Sephadex LH20 with methanol as the eluent. FTIR spectra were recorded using a Shimadzu IRAffinity-1, ATR PIKE Technologies model GladiAT (cm^−1^) (Cottonwood, WI, USA). Optical rotations were measured using an Anton-Paar MCP 300 polarimeter (Graz, Austria). ^1^H, ^13^C and ^19^F NMR spectra were recorded using a Bruker Avance III (400 MHz, 100.6 MHz, and 376 MHz, respectively, Billerica, MA, USA) on the NMR Platform of the Jean Barriol Institute (Université de Lorraine, Nancy, France). For the complete assignment of ^1^H and ^13^C signals, two-dimensional ^1^H, ^1^H COSY and ^1^H, ^13^C correlation spectra were recorded. Chemical shifts (δ) are given in parts per million relative to the solvent residual peak. The following abbreviations are used for the multiplicity of NMR signals: s = singlet, d = doublet, t = triplet, q = quadruplet, m = multiplet, b = broad signal, and app = apparent multiplicity. Atom numbering used in NMR attribution signals is provided on copies of NMR spectra (see ESI). The given *J* values refer to apparent multiplicities and do not represent the true coupling constants. High resolution ESI-MS spectra were recorded using a Bruker Daltonics microTOFQ apparatus provided by the mass spectrometry MassLor platform of Université de Lorraine. UV-vis spectra were recorded using a PerkinElmer Lambda1050 spectrophotometer (Waltham, MA, USA), and room temperature fluorescence emission spectra were recorded using a Fluorolog-3 spectrofluorometer from Horiba Scientific (Kyoto, Japan). All the spectroscopic measurements were performed using the PhotoNS spectroscopic platform of the L2CM Laboratory. No-carrier-added fluoride-18 was produced via the ^18^O (p,n)^18^F nuclear reaction using a PET Trace cyclotron (GE) or Cyclone-18/9 cyclotron (18 MeV proton beam, IBA). For the PET Trace cyclotron, the bombardment was performed at 10 µA for 5 min to provide about 2 GBq of fluoride-18 delivered as a solution in ^18^O-enriched water (1.6 mL). In the case of Cyclone-18/9 cyclotron, the bombardment was performed at 6.2 µA for 14 min to provide about 17 GBq of fluoride-18 delivered as a solution in ^18^O-enriched water (2.0 mL) Radiosynthesis was performed using an AIO module from Trasis^®^ or a TracerLab FxFN from General Electric^®^ (GE, Boston, MA, USA). Analytical High-Performance Liquid Chromatography (HPLC) analyses were performed using a Waters system (2695eb pump, auto sampler injector and 2998 PDA detector) coupled to a radioHPLC detector (Herm LB500 with NaI from Berthold, Bad Wildbad, Germany) controlled by the Empower Software (Orlando, FL, USA) or using a Waters Alliance 2690 (UV spectrophotometer (Photodiode Array Detector, Waters 996 (Waters)) and a Berthold LB509 radioactivity detector). The analyses were performed using a Luna PFP column (5 μm, 150 × 4.6 mm) from Phenomenex (Torrance, CA, USA) with ACN/H_2_O (*v*/*v* 60/40) eluant and 0.1% of TFA at 1 mL/min (Waters system) or at 1.5 mL/min (Waters Alliance 2690). U.V. detection at λ = 650 nm.

### 3.2. Chemistry

Compounds **4**, **5** [32] and **21** [50] were prepared according to previously described methods (see Appendix A).

#### 3.2.1. 3,6-anhydro-2-deoxy-4,5-O-(1-methylethylidene)-7-fluoro-D-ribo-heptanoic acid methyl ester **6**

To a solution of **5** (100 mg, 0.40 mmol) in diglyme (3 mL), 110 µL of DAST (2 eq., 0.81 mmol) were added dropwise at 0 °C under an inert atmosphere. The reaction mixture was stirred for 30 min at 0 °C and 1 h 30 at 110 °C. After cooling at room temperature, the mixture was neutralized by the addition of a saturated solution of NaHCO_3_ (10 mL), and the solvent was removed under vacuum. The residue was solubilized in water, the aqueous layer was extracted with CH_2_Cl_2_ (2 × 20 mL), and the organic layer was dried over MgSO_4_. The solvent was removed under vacuum, and the crude product was purified using a flash chromatography on silica gel (eluent: cyclohexane/EtOAc 100/0 to 60/40) to afford compound **6**. Yield: 85% as a colorless oil, *R_f_ =* 0.85 (Cycl/EtOAc: 6/4), [α]_D_ = −11.3 (c = 0.10, CHCl_3_). IR (cm^−1^): ν = 2988, 2955, 1734, 1437, 1381, 1307, 1267. ^1^H NMR (CDCl_3_, 400 MHz): δ (ppm) = 1.34 (s, 3H, *CH_3_*), 1.49 (s, 3H, *CH_3_*), 2.72 (dd, 1H, *J*_2a,2b_ = 16.5 Hz, *J*_2a,3_ = 7.0 Hz, *H-2a*), 2.78 (dd, 1H, *J*_2b,3_ = 7.0 Hz, *H-2b*), 3.71 (s, 3H, *OCH*_3_), 4.19 (app dt, 1H, *J*_6,F_ = 33.0 Hz, *J*_6,7a_ = *J*_6,7b_ = 3.0 Hz, *H-6*), 4.40–4.46 (m, 1H, *H-3),* 4.46 (ddd, 1H, *J*_7a,F_ = 47.5 Hz, *J*_7b,7a_ = 10.5 Hz, *J*_7b,6_ = 3.5 Hz, *H-7a*), 4.58 (ddd, 1H, *J*_7b,F_ = 47.5 Hz, *J*_7b,7a_ = 10.5 Hz, *J*_7b,6_ = 3.0 Hz, *H-7b*), 4.80 (dd, 1H, *J*_4,5_ = 6.0 Hz, *J*_4,3_ = 4.0 Hz, *H-4*), 4.85 (app brd, 1H, *H-5*). ^13^C NMR (CDCl_3_, 100.6 MHz): δ (ppm) = 25.1 (*CH_3_*), 26.3 (*CH_3_*), 36.4 (*C-2*), 51.9 (*OCH_3_*), 78.6 (d, *J_C3-F_* = 2 Hz, *C-3*), 81.6 (*C-4*), 82.4 (d, *J_C5-F_* = 8 Hz, C*-5*), 82.5 (d, *J_C6-F_* = 19 Hz, *C-6*), 85.1 (d, *J_C7-F_* = 172 Hz, *C-7*), 112.9 (*C*(CH_3_)_2_), 171.6 (*C=O*). ^19^F NMR (CDCl_3_, 376 MHz): δ (ppm) = −229.2. ESI-HRMS [M + Na]^+^ m/z = 271.0952 (calculated for C_11_H_18_FNaO_5_: 271.0958).

#### 3.2.2. 3,6-anhydro-2-deoxy-4,5-O-(1-methylethylidene)-7-O-(tert-butyldiphenylsilyl)-D-ribo-heptanoic acid methyl ester **7**

To a solution of compound **5** (1 g, 4.05 mmol) in dry DMF (10 mL), a solution of 1.58 mL of *tert*-butyldiphenylsilyl chloride (1.5 eq., 6.07 mmol) and 496 mg of imidazole (1.8 eq., 7.29 mmol) in dry DMF (3mL) was added dropwise at 0 °C under an inert atmosphere. The reaction was stirred overnight at room temperature, and the mixture was then diluted with water (50 mL) and extracted with EtOAc (3 × 80 mL). The organic layer was dried over MgSO_4_ and filtered, and the solvent was removed under vacuum. The crude product was purified using a flash chromatography on silica gel (eluent: cyclohexane/EtOAc 100/0 to 90/10) to afford compound **7**. Yield: 95% as colorless oil, *R_f_ =* 0.47 (Cycl/EtOAc: 8/2), [α]_D_ = +0.4 (c = 0.36, CHCl_3_). IR (cm^−1^): ν = 2928, 1742, 1425, 1351, 1210. ^1^H NMR (CDCl_3_, 400 MHz): δ (ppm) = 1.06 (s, 9H, Si*-tert-butyl*), 1.35 (s, 3H, C*H*_3_), 1.49 (s, 3H, C*H*_3_), 2.70 (dd, 1H, *J*_2a,2b_ = 16.5 Hz, *J*_2a,3_ = 7.0 Hz, *H*-2a), 2.76 (dd, 1H, *J*_2b,3_ = 7.0 Hz, *H*-2b), 3.67 (dd, 1H, *J*_7a,7b_ = 11.0 Hz, *J*_7a,6_ = 4.0 Hz, *H*-7a), 3.71 (s, 3H, COOC*H*_3_), 3.77 (dd, 1H, *J*_7b,6_ = 4.0 Hz, *H*-7b), 4.11 (app br t, 1H, *H-*6), 4.59 (app td, 1H, *J*_3,2a_ = *J*_3,2b_ 7.0 Hz, *J*_3,4_ = 4.0 Hz, *H-*3), 4.80 (dd, 1H, *J*_4,5_ = 6.0 Hz, *H-*4), 4.86 (br d, 1H, *H*-5), 7.36–7.46 (m, 6H, *H_Ar_*), 7.64–7.68 (m, 4H, *H_Ar_*). ^13^C NMR (CDCl_3_, 100.6 MHz): δ (ppm) = 19.2 (*C-Si*), 25.2 (*CH_3_*), 26.4 (*CH_3_*), 27.0 *(*Si*-tert-butyl*), 34.9 (*C-2*), 51.8 (*OCH_3_*), 65.4 (*C-7*), 78.6 (*C-3*), 82.0 (*C-4*), 83.4 *(C-5)*, 84.3 (*C-6*), 112.5 (*C*(CH_3_)_2_), 128.0 (4*C_Ar_*), 130.0 (*C_Ar_*), 130.0 (*C_Ar_*), 132.9 (*Cq_Ar_*), 133.0 (*Cq_Ar_*), 135.8 (4*C_Ar_*), 171.8 (*C=O*). ESI-HRMS [M + K]^+^ m/z = 523.1877 (calculated for C_27_H_36_KO_6_Si: 523.1913).

#### 3.2.3. 3,6-anhydro-2-deoxy-4,5-O-(1-methylethylidene)-7-fluoro-D-ribo-1-hydroxyl-heptane **8**

To a solution of **7** (92 mg, 0.37 mmol, 1.0 eq.) in dry THF (8 mL), 42 mg of LiAlH_4_ (1.11 mmol, 3.0 eq.) was added at 0 °C under an inert atmosphere, and the mixture was stirred at room temperature for 3 h. The reaction was quenched with the addition of water, and the mixture was filtered off using a Celite^®^ pad. The organic solvent was removed under reduced pressure, and the aqueous layer was extracted with CH_2_Cl_2_ (3 × 50 mL). The combined organic layers were dried over MgSO_4_, and the solvent was removed under vacuum. The crude product was purified using a flash chromatography on silica gel (eluent: cyclohexane/EtOAc 100/0 to 50/50) to afford compound **8**. Yield: 82% as yellowish oil, *R_f_ =* 0.26 (Cycl/EtOAc: 6/4), [α]_D_ = −3.4 (c = 0.05, CHCl_3_). IR (cm^−1^): ν = 3420, 2932, 1456, 1373, 1269, 1234, 1209. ^1^H NMR (CDCl_3_, 400 MHz): δ (ppm) = 1.35 (s, 3H, *CH_3_*), 1.51 (s, 3H, *CH_3_*), 1.89–2.06 (m, 2H, *H-2*), 3.76–3.86 (m, 2H, *H-1*), 4.14–4.21 (m, 1H, *H-3*), 4.21 (app dt, 1H, *J*_6,F_ = 27.0 Hz, *J*_6,7_ = *J*_6,5_ = 2.5 Hz, *H*-6), 4.44–4.62 (m, 2H, *H-7),* 4.72 (dd, 1H, *J*_4,5_ = 6.0 Hz, *J*_4,3_ = 3.5 Hz, *H-4*), 4.85 (br dt, 1H, *H-5*). ^13^C NMR (CDCl_3_, 100.6 MHz): δ (ppm) = 25.1 (*CH_3_*), 26.4 (*CH_3_*), 32.2 (*C-2*), 60.9 (*C-1*), 81.6 (d, *J_C3-F_* = 2 Hz, *C-3*), 82.3 (d, *J_C5-F_* = 6.0 Hz, C*-5*), 82.4 (*C-4*), 82.7 (d, *J_C6-F_* = 17.0 Hz, *C-6*), 85.3 (d, *J_C7-F_* = 172.0 Hz, *C-7*), 112.8 (*C*(CH_3_)_2_). ^19^F NMR (CDCl_3_, 376 MHz): δ (ppm) = -228.9. ESI-HRMS [M + Na]^+^ m/z = 243.1081 (calculated for C_10_H_17_FNaO_4_: 243.1003).

#### 3.2.4. 3,6-anhydro-2-deoxy-4,5-O-(1-methylethylidene)-7-O-(tert-butyldiphenylsilyl)-D-ribo-1-hydroxyl-heptane **9**

Prepared from **7** following the procedure described for **8**. Yield: 76% as yellowish oil, *R_f_ =* 0.18 (Cycl/EtOAc: 8/2), [α]_D_ = +12.68 (c = 0.07, CHCl_3_)**.** IR (cm^−1^): ν = 3442, 2929, 2856, 1472, 1427, 1380, 1207, 1111. ^1^H NMR (CDCl_3_, 400 MHz): δ (ppm) = 1.06 (s, 9H, *Si-tert-butyl*), 1.36 (CH_3_), 1.51 (CH_3_), 1.85–1.94 (m, 1H, *H-2a*), 1.98–2.08 (m, 1H, *H-2b*), 2.17 (bs, OH), 3.70 (app dd, 1H, *J*_7a,7b_ = 11.0 Hz, *J*_7a,6_ = 4.0 Hz, *H-7a*), 3.77–3.86 (m, 1H, *J*_7b,6_ = 4.0 Hz, *H-7b*), 3.78–3.87 (m, 2H, *H-1*), 4.13 (app t, 1H, *H-6*), 4.29 (m, 1H, *H-3*), 4.70 (dd, 1H, *J*_4,5_ = 6.0 Hz, *J*_4,3_ = 4.0 Hz, *H-4*), 4.86 (dd, 1H, *J*_5,6_ = 1.0 Hz, *H-5*), 7.36–7.47 (m, 6H, *H_A_*_r_), 7.63–7.68 (m, 4H, *H_A_*_r_). ^13^C NMR (CDCl_3_, 100.6 MHz): 19.2 (C, Si-*C*), 25.2 (*CH_3_*), 26.5 (*CH_3_*), 27.0 (3C, Si-C*(CH_3_)_3_*), 32.3 (*C-2*), 61.2 (*C-1*), 65.4 (*C-7*), 81.8 (*C-3*), 82.7 (*C-4*), 83.2 (*C-5*), 84.5 (*C-6*), 112.5 (*C*(CH_3_)_2_), 128.0 (4*C_Ar_*), 130.0 (*C_Ar_*), 130.1 (*C_Ar_*), 133.0 (*Cq_Ar_*), 133.1 (*Cq_Ar_*), 135.6 (2*C_Ar_*), 135.7 (2*C_Ar_*). ESI-HRMS [M + Na]^+^ m/z = 479.2236 (calculated for C_26_H_36_NaO_5_Si: 479.2224).

#### 3.2.5. 3,6-anhydro-2-deoxy-4,5-O-(1-methylethylidene)-7-fluoro-D-ribo-(prop-2-yne-1-yloxy)heptane **10**

To a suspension of NaH 60% in mineral oil (150 mg, 3.91 mmol, 2.0 eq.) in dry DMF (4 mL), a solution of 431 mg of **8** (0.48 mmol, 1.0 eq.) in dry DMF (4 mL) was added at 0 °C under an inert atmosphere. After 1 h, 654 µL of propargyl bromide with 80% in toluen (5.88 mmol, 3.0 eq.) was added, and the mixture was stirred at room temperature for 16 h. The reaction was quenched with the addition of an aqueous saturated solution of NH_4_Cl. The solvent was removed under reduced pressure, and the resultant residue was dissolved in water. The aqueous layer was extracted with CH_2_Cl_2_ (3 × 50 mL). The organic layer was dried over MgSO_4_, and the solvent was removed under vacuum. The crude product was purified using a flash chromatography on silica gel (eluent: cyclohexane/EtOAc 100/0 to 50/50) to afford compound **10** Yield: 95% as yellowish oil. *R_f_ =* 0.77(Cycl/EtOAc: 6/4), [α]_D_ = −5.86 (c = 0.45, CHCl_3_). IR (cm^−1^): ν = 3265, 2986, 2122, 1718, 1375, 1232, 1209. ^1^H NMR (CDCl_3_, 400 MHz): δ (ppm) = 1.34 (s, 3H, *CH_3_*), 1.49 (s, 3H, *CH_3_*), 1.98 (app q, 2H, *J*_2,3_ = *J*_2,1_ = 6.5 Hz, *H-2*), 2.41 (t, 1H, *J*_10,8_ = 2.5 Hz, *H-10*), 3.62–3.72 (m, 2H, *H-1*), 4.10–4.15 (m, 1H, *H-3*), 4.15 (d, 2H, *H-8*), 4.14–4.23 (m, 1H, *H-6*), 4.42–4.59 (m, 2H, *H-7),* 4.68 (dd, 1H, *J*_4,5_ = 6.0 Hz, *J*_4,3_ = 4.0 Hz, *H-4*), 4.82 (br d, 1H, *H-5*). ^13^C NMR (CDCl_3_, 100.6 MHz): δ (ppm) = 25.2 (*CH_3_*), 26.4 (*CH_3_*), 29.7 (*C-2*), 58.3 (*C-8*), 67.3 (*C-1*), 74.3 (*C-9*), 79.5 (d, *J_C3-F_* = 2 Hz, *C-3*), 80.0 (*C-10*), 82.0 (*C-4*), 82.4 (d, *J_C5-F_* = 6.0 Hz, C*-5*), 82.4 (d, *J_C6-F_* = 18.0 Hz, *C-6*), 84.9 (d, *J_C7-F_* = 172.0 Hz, *C-7*), 112.7 (*C*(CH_3_)_2_). ^19^F NMR (CDCl_3_, 376 MHz): δ (ppm) = −228.7. ESI-HRMS [M + Na]^+^ m/z = 281.1137 (calculated for C_13_H_19_FNaO_4_: 281.1160).

#### 3.2.6. 3,6-anhydro-2-deoxy-4,5-O-(1-methylethylidene)-7-O-(tert-butyldiphenylsilyl)-D-ribo-(prop-2-yne-1-yloxy)heptane **11**

To a suspension of NaH 60% in mineral oil (27 mg, 0.66 mmol, 1.2 eq.) in dry THF (1 mL), a solution of 250 mg of **9** (0.55 mmol, 1.0 eq.) in dry THF (2 mL) was added at 0 °C under an inert atmosphere. After 10 min, 183 µL of propargyl bromide 80% in toluene (1.64 mmol, 3.0 eq.) was added, and the mixture was stirred at room temperature for 6 h. The reaction was quenched with the addition of an aqueous saturated solution of NH_4_Cl. The solvent was removed under reduced pressure, and the residue was dissolved in water. The aqueous layer was extracted with CH_2_Cl_2_ (3 × 50 mL). The organic layer was dried over MgSO_4_, and the solvent was removed under vacuum. The crude product was purified using a flash chromatography on silica gel (eluent: cyclohexane/EtOAc 100/0 to 60/40) to afford compound **11**. Yield: 70% as colorless oil, *R_f_ =* 0.77 (Cycl/EtOAc: 8/2), [α]_D_ = +9.29 (c = 0.10, CHCl_3_). IR (cm^−1^): ν = 2926, 2855, 2116, 1670, 1472, 1427, 1361, 1103. ^1^H NMR (CDCl_3_, 400 MHz): δ (ppm) = 1.06 (s, 9H, *Si-tert-butyl*), 1.36 (CH_3_), 1.51 (CH_3_), 1.99 (m, 2H, *H-2*), 2.38 (t, 1H, *J*_10–8_ = 2.5 Hz, *H-10*), 3.61–3.78 (m, 4H, *H-1* and *H-7*), 4.09 (app t, 1H, *H-6*), 4.14 (d, 2H, *J* = 2.5 Hz, *H-8*), 4.24–4.29 (td, 1H, *J*_4,3_ = 4.0 Hz, *H-*3), 4.67 (dd, 1H, *J*_4,5_ = 6.0 Hz, *H-4*), 4.85 (dd, 1H, *J*_5,6_ = 1.0 Hz, *H-5*), 7.35–7.44 (m, 6H, *H_A_*_r_), 7.66–7.70 (m, 4H, *H_A_*_r_). ^13^C NMR (CDCl_3_, 100.6 MHz): 19.3 (C, Si-*C*), 25.3 (*CH_3_*), 26.5 (*CH_3_*), 27.0 (3C, Si-C*(CH_3_)_3_*), 29.9 (*C-2*), 58.2 (*C-8*), 65.3 (*C-7*), 67.5 (*C-1*), 74.3 (*C-10*), 79.4 (*C-3*) 80.1 (*C-9*), 82.4 (*C-4*), 83.4 (*C-5*), 84.3 (*C-6*), 112.3 (*C*(CH_3_)_2_), 127.9 (2*C_Ar_*), 128.0 (2*C_Ar_*), 129.9 (*C_Ar_*), 129.8 (*C_Ar_*), 133.1 (*Cq_Ar_*), 133.2 (*Cq_Ar_*), 135.0 (2*C_Ar_*), 135.7 (2*C_Ar_*). ESI-HRMS [M + Na]^+^ m/z = 517.2482 (calculated for C_29_H_38_NaO_5_Si: 517.2381).

#### 3.2.7. 3,6-anhydro-2-deoxy-4,5-hydroxy-7-fluoro-D-ribo-(prop-2-yne-1-yloxy)heptane **12**

A solution of **10** (500 mg, 1.94 mmol) in a mixture of TFA/H_2_O 6/4 (*v*/*v*) (20 mL) was stirred for 3 h at room temperature. The solvent was removed under vacuum, and the crude residue was purified using a column chromatography on silica gel (eluent: EtOAc/MeOH 100/0 to 90/10) to afford compound **12**. Yield: 84% colorless oil, *R_f_ =* 0.13 (Cycl/EtOAc: 7/3), [α]_D_ = −2.6 (c = 0.32, CHCl_3_). IR (cm^−1^): ν = 3385, 3283 2951, 289, 2116, 1717, 1356. ^1^H NMR (CDCl_3_, 400 MHz): δ (ppm) = 1.93–2.02 (m, 1H, *H-2a*), 2.08–2.18 (m, 1H, *H-2b*), 2.47 (t, 1H, *J*_10,8_ = 2.5 Hz, *H-10*), 3.56 (ddd, 1H, *J*_1a,1b_ = 10.5 Hz, *J*_1a,2b_ = 9.5 Hz, *J*_1a,2a_ = 2.5 Hz, *H-1a*), 3.72 (ddd, 1H, *J*_1b,2b_ = 3.5 Hz, *J*_1b,2a_ = 5.0 Hz, *H-1b*), 3.94 (dddd, 1H, *J*_6,F_ = 27.0 Hz, *J*_6,5_ = 7.0 Hz, *J*_6,7a_ = 4.0 Hz, *J*_6,7b_ = 2.5 Hz, *H-6*), 4.08–4.13 (m, 1H, *H-4*), 4.14–4.19 (m, 1H, *H-3*), 4.15 (dd, 1H, *J*_8a,8b_ = 12.5 Hz, *H-8a*), 4.20 (dd, 1H, *H-8b*), 4.23 (dd, 1H, *J*_5,6_ = 7.0 Hz, *J*_5,4_ = 5.0 Hz *H-5*), 4.51 (ddd, 1H, *J*_7a,F_ = 47.5 Hz, *J*_7a,7b_ = 10.5 Hz, *H-7a*), 4.63 (ddd, 1H, *J*_7b,F_ = 48.0 Hz, *H-7b*). ^13^C NMR (CDCl_3_, 100.6 MHz): 29.9 (*C-2*), 58.6 (*C-8*), 66.7 (*C-1*), 72.1 (d, *J_C3-F_* = 1.5 Hz, *C-3*), 72.3 (d, *J_C5-F_* = 7.5 Hz, *C-5*), 75.3 (*C-10*), 79.0 (*C-9*), 80.6 (d, *J_C6-F_* = 18.0 Hz, *C-6*), 80.7 (*C-4*), 83.1 (d, *J_C7-F_* = 172.0 Hz, *C-7*). ^19^F NMR (CDCl_3_, 376 MHz): δ (ppm) = −232.1. ESI-HRMS [M + Na]^+^ m/z = 241.0818 (calculated for C_10_H_15_FNaO_4_: 241.0847).

#### 3.2.8. Compound **13**

A solution of **11** (70 mg, 0.14 mmol, 1.0 eq.) in AcOH 80% in water (1.5 mL) was stirred at 80 °C for 4 h. The mixture was cooled at 0 °C, and the reaction wash quenched with the addition of water (5 mL) and solid NaHCO_3_ until pH = 7. The aqueous layer was extracted with CH_2_Cl_2_ (3 × 10 mL), the combined organic layers were dried over MgSO_4_, and the solvent was removed under vacuum. The crude product was purified using a flash chromatography on silica gel (eluent: cyclohexane/EtOAc 100/0 to 60/40) to afford compound **13**. Yield: 54% colorless oil, *R_f_ =* 0.13 (Cycl/EtOAc: 8/2), [α]_D_ = +33.25 (c = 0.04, CHCl_3_). IR (cm^−1^): ν = 3379, 3302, 2927, 2852, 2119, 1666, 1462, 1103. ^1^H NMR (CDCl_3_, 400 MHz): δ (ppm) = 1.06 (s, 9H, *Si-tert-butyl*), 1.94–2.02 (m, 1H, *H-2a*), 2.04–2.15 (m, 1H, *H-2b*), 2.44 (t, 1H, *J*_10–8_ = 2.5 Hz, *H-10*), 3.58 (app td, 1H, *J*_1a,1b_ = *J*_1a,2a_ = 9.5 Hz, *J*_1a,2b_ = 3.5 Hz, *H-1a*), 3.69–3.76 (m, 1H, *H-1b*), 3.81 (dd, 1H, *J*_7a,7b_ = 11.5 Hz, *J*_7a,6_ = 4.5 Hz, *H-7a*), 3.84–3.90 (m, 2H, *H-6* and *H-7b*), 4.09–4.14 (m, 1H, *H-3*), 4.14–4.18 (m, 3H, *H-*4 and *H-*8), 4.43 (dd, 1H, *J* = 6.5 Hz, *J* = 5.0 Hz, *H-5*), 7.35–7.45 (m, 6H, *H_A_*_r_), 7.66–7.73 (m, 4H, *H_A_*_r_). ^13^C NMR (CDCl_3_, 100.6 MHz): 19.4 (C, Si-*C*), 27.0 (3C, Si-C*(CH_3_)_3_*), 30.0 (*C-2*), 58.5 (*C-8*), 64.4 (C-7), 66.9 (*C-1*), 72.7 (*C-4*), 73.6 (*C-5*), 75.0 (*C-10*), 79.3 (*C-9*), 80.1 (*C-3*), 82.2 (*C-6*), 127.8 (2*C_Ar_*), 127.9 (2*C_Ar_*), 129.8 (*C_Ar_*), 129.9 (*C_Ar_*), 133.5 (*Cq_Ar_*), 133.6 (*Cq_Ar_*), 135.8 (2*C_Ar_*), 135.8 (2*C_Ar_*). ESI-HRMS [M + Na]^+^ m/z = 477.2088 (calculated for C_26_H_34_NaO_5_Si: 477.2068).

#### 3.2.9. 3,6-anhydro-2-deoxy-4,5-O-(3-(triisopropylsilyl)prop-2-yne)-7-fluoro-D-ribo-(prop-2-yne-1-yloxy)heptane **14**

To a suspension of NaH 60% in mineral oil (88 mg, 2.20 mmol, 3.0 eq.) in dry DMF (1 mL), a solution of 160 mg of **12** (0.73 mmol, 1.0 eq.) in dry DMF (3 mL) was added at 0 °C under an inert atmosphere. After 45 min, 1.2 g of 3-bromo-1-(triisopropylsilyl)-1-propyne (4.38 mmol, 6.0 eq.) was added, and the mixture was stirred at room temperature for 5 h. The reaction was quenched with the addition of an aqueous saturated solution of NH_4_Cl. The solvent was removed under vacuum, and the residue was dissolved in water. The aqueous layer was extracted with CH_2_Cl_2_ (3 × 50 mL). The organic layer was dried over MgSO_4_, and the solvent was removed under vacuum. The crude product was purified using a flash chromatography on silica gel (eluent: cyclohexane/EtOAc 100/0 to 50/50) to afford compound **12**. Yield: 94% as yellowish oil, *R_f_ =* 0.59 (Cycl/EtOAc: 8/2), [α]_D_ = +24.2 (c = 0.10, CHCl_3_). IR (cm^−1^): ν = 3298, 2941, 2864, 1717, 1458, 1383, 1227. ^1^H NMR (CDCl_3_, 400 MHz): δ (ppm) = 1.07 (br s, 42H, Si-*isopropyl* and Si-*isopropyl*), 1.97 (app q, 2H, *J*_2,1_ = *J*_2,3_ = 6.5 Hz, *H-2*), 2.39 (t, 1H, *J*_16,14_ = 2.5 Hz, *H-16*), 3.60–3.69 (m, 2H, *H-1*), 4.03–4.12 (m, 1H, *H-6*), 4.10 (dd, 1H, *J*_14a,14b_ = 15.0 Hz, *H-14a*), 4.15 (dd, 1H, *H-14b*), 4.18–4.23 (td, 1H, *J*_3,4_ = 4.0 Hz, *H-3*), 4.28 (app t, 1H, *J*_4,3_ = *J*_4,5_ = 4.0 Hz, *H-4*), 4.30–4.34 (m, 1H, *H-5*), 4.35 (d, 2H, *J* = 1.0 Hz, *H-8* or *H-11*), 4.45 (ddd, 1H, *J*_7a,F_ = 47.0 Hz, *J*_7a,7b_ = 10.0 Hz, *J*_7a,6_ = 4.0 Hz, *H-7a*), 4.45 (d, 2H, *J* = 1.0 Hz, *H-8* or *H-11*), 4.56 (ddd, 1H, *J*_7b,F_ = 48.0 Hz, *J*_7b,6_ = 2.5 Hz, *H-7b*). ^13^C NMR (CDCl_3_, 100.6 MHz): 11.3 (3C, Si-*C*H-(CH_3_)_2_), 11.3 (3C, Si-*C*H-(CH_3_)_2_), 18.7 (6C, Si-CH-(*C*H_3_)_2_), 18.7 (6C, Si-CH-(*C*H_3_)_2_), 30.1 (*C-2*), 58.2 (*C-14*), 58.7 (*C-8* or *C-11*), 59.2 (*C-8* or *C-11*), 67.0 (*C-1*), 76.0 (*C-4*), 77.8 (*C-3*), 77.9 (d, *J_C5-F_* = 6.0 Hz, *C-5*), 78.7 (d, *J_C6-F_* = 18.5 Hz, *C-6*), 80.1 (*C-15*), 82.8 (d, *J_C7-F_* = 173.5 Hz, *C-7*), 88.6 (*C-10* or *C-13*), 89.0 (*C-10* or *C-13*), 102.5 (*C-9* or *C-12*), 103.2 (*C-9* or *C-12*). ^19^F NMR (CDCl_3_, 376 MHz): δ (ppm) = −231.0. ESI-HRMS [M + K]^+^ m/z = 645.3541 (calculated for C_34_H_59_FKO_4_Si_2_: 645.3567).

#### 3.2.10. 3,6-anhydro-2-deoxy-4,5-O-(3-(triisopropylsilyl)prop-2-yne)-7-O-(tert-butyldiphenylsilyl)-D-ribo-(prop-2-yne-1-yloxy)heptane **15**

To a suspension of NaH 60% in mineral oil (12 mg, 0.29 mmol, 2.2 eq.) in dry THF (1 mL), a solution of 60 mg of **13** (0.13 mmol, 1.0 eq.) in dry THF (1 mL) was added at 0 °C under an inert atmosphere. After 10 min, 218 mg of 3-bromo-1-(triisopropylsilyl)-1-propyne (0.79 mmol, 6.0 eq.) was added, and the mixture was stirred at room temperature for 5 h. The reaction was quenched with the addition of an aqueous saturated solution of NH_4_Cl. The organic solvent was removed under reduced pressure. The aqueous layer was extracted with CH_2_Cl_2_ (3 × 50 mL). The organic layer was dried over MgSO_4_, and the solvent was removed under vacuum. The crude product was purified using a flash chromatography on silica gel (eluent: cyclohexane/EtOAc 100/0 to 50/50) to afford compound **15**. Yield: 78% as colorless oil, *R_f_ =* 0.59 (Cycl/EtOAc: 8/2), [α]_D_ = +20.34 (c = 0.09, CHCl_3_). IR (cm^−1^): ν = 2939, 2889, 2862, 2253, 2167, 2115, 1462, 1103. ^1^H NMR (CDCl_3_, 400 MHz): δ (ppm) = 1.05 (s, 21H, *Si-isopropyl*), 1.06 (s, 9H, *Si-tert-butyl*), 1.07 (s, 21H, *Si-isopropyl*), 1.93–2.01 (m, 2H, *H-2*), 2.34 (t, 1H, *J*_10–8_ = 2.5 Hz, *H-16*), 3.59–3.72 (m, 3H, *H-1* and *H-7a*), 3.78 (dd, 1H, *J*_7b,7a_ = 11.0 Hz, *J*_7b,6_ = 4.0 Hz, *H-7a*), 4.01–4.06 (m, 1H, *H6*)*,* 4.11 (bt, 2H, H-14), 4.15–4.21 (m, 1H, *H-3*), 4.24–4.33 (m, 2H, *H-4* and *H-5*), 4.26 (d, 1H, *J* = 16.0 Hz, Hz, *H-8a* or *H-11a)*, 4.33 (d, 1H, *J* = 16.0 Hz, Hz, *H-8b* or *H-11b)*, 4.42 (d, 1H, *J* = 16.5 Hz, Hz, *H-8b or H-11b)*, 4.49 (d, 1H, *J* = 16.5 Hz, Hz, *H-8b or H-11b)*, 7.34–7.44 (m, 6H, *H_A_*_r_), 7.66–7.71 (m, 4H, *H_A_*_r_). ^13^C NMR (CDCl_3_, 100.6 MHz): 11.3 (3C, Si-*C*H-(CH_3_)_2_), 11.3 (3C, Si-*C*H-(CH_3_)_2_), 18.7 (6C, Si-CH-(*C*H_3_)_2_), 18.7 (6C, Si-CH-(*C*H_3_)_2_), 19.4 (C, Si-*C*), 27.0 (3C, Si-C*(CH_3_)_3_*), 30.2 (*C-2*), 58.2 (*C-14*), 58.6 (*C-8* or *C-11*), 59.2 (*C-8* or *C-11*), 64.7 (C-7), 67.2 (*C-1*), 74.1 (*C-16*), 77.4 (*C-3* and *C-4*), 79.4 (*C-5*), 80.2 (*C-15*), 81.2 (*C-6*), 87.9 (*C-10* or *C-13*), 88.0 (*C-10* or *C-13*), 103.3 (*C-9* or *C-12*), 103.6 (*C-9* or *C-12*), 127.8 (2*C_Ar_*), 127.8 (2*C_Ar_*), 129.8 (*C_Ar_*), 129.8 (*C_Ar_*), 133.5 (*Cq_Ar_*), 133.7 (*Cq_Ar_*), 135.8 (2*C_Ar_*), 135.8 (2*C_Ar_*). ESI-HRMS [M + H]^+^ m/z = 843.5243 (calculated for C_50_H_79_O_5_Si_3_: 843.5235).

#### 3.2.11. 3H-Indolium,2-[5-(1,3-dihydro-1,3,3-trimethyl-2H-indol-2-ylidene)-1,3-pentadien-1-yl]-3,3-dimethyl-1-[6-oxo-6-(3-azidopropylamino)hexyl]-iodonium salt **16**

To a solution of commercial *N*-hydroxysuccinimide cyanine **5** (90 mg, 0.13 mmol, 1.0 eq.) in DMF (2 mL), 35 mg of 3-azidopropyl-1-amine hydrochloride [51] (0.26 mmol, 2.0 eq.) and 40 µL of DIPEA (0.39 mmol, 3.0 eq.) were added, and the mixture was stirred at room temperature for 6 h. The reaction mixture was then diluted with CH_2_Cl_2_ (20 mL), and the organic layer was washed with water (3 × 10 mL). The organic layer was dried over MgSO_4_, and the solvent was removed under vacuum. The crude product was purified using a flash chromatography on silica gel (eluent: DCM/MeOH 100/0 to 90/10) to afford compound **16**. Yield: 69% as a blue solid, *R_f_ =* 0.61 (DCM/MeOH: 95/5). IR (cm^−1^): ν = 2928, 2095, 1649, 1479, 1452, 1369, 1335, 1219. ^1^H NMR (CDCl_3_, 400 MHz): δ (ppm) = 1.52–1.64 (m, 2H, *H-6*), 1.69 (br s, 12H, 4 *CH_3_*), 1.76–1.92 (m, 6H, *H-2, H-5, H-7)*, 2.44 (br t, 2H, *J*_4,5_ = 6.5 Hz, *H-4*), 3.29–3.36 (m, 2H, *H-3*), 3.38 (t, 2H, *J*_1,2_ = 7.0 Hz, *H-1*) 3.59 (s, 3H, *N-CH_3_*), 4.12 (br t, 2H, *J*_8,7_ = 6.5 Hz, *H-8*), 6.31 (d, 1H, *J* = 14.0 Hz, *H-11* or *H-15*), 6.71 (d, 1H, *H-11* or *H-15*), 6.99–7.08 (m, 1H, *H-13*), 7.06 (d, 1H, *J* = 7.5 Hz, *H_Ar_*), 7.13 (d, 1H, *J* = 7.5 Hz, *H_Ar_*), 7.18–7.25 (m, 2H, *H_Ar_*), 7.32–7.41 (m, 4H, *H_Ar_*), 7.82 (app t, 1H, *J* = 10.5 Hz, *H-12* or *H-14*), 7.86 (app t, 1H, *J* = 10.5 Hz, *H-12* or *H-14*), 8.59 (br s, 1H, *NH*). ^13^C NMR (CDCl_3_, 100.6 MHz): δ (ppm) = 25.2 (*C-2, C-5* or *C-7*), 26.4 (*C-6*), 27.0 (*C-2, C-5* or *C-7*), 28.2 (2 *CH_3_)*, 28.3 (2 *CH_3_)*, 29.1 (*C-2, C-5* or *C-7*), 31.5 (N-*CH_3_*), 36.0 (*C-4*), 36.9 (*C-3*), 44.9 (*C-8*), 49.0 (*C-9* or *C-17*), 49.5 (*C-1*), 49.8 (*C-9* or *C-17*), 103.8 (*C-11* or *C-15*), 105.5 (*C-11* or *C-15*), 110.2 (*C_Ar_*), 111.2 (*C_Ar_*), 122.2 (*C_Ar_*), 122.3 (*C_Ar_*), 125.6 (*C_Ar_*), 127.1 (*C-13*), 128.8 (*C_Ar_*), 129.0 (*C_Ar_*), 140.7 (*Cq_Ar_*), 141.3 (*Cq_Ar_*), 142.0 (*Cq_Ar_*), 143.0 (*Cq_Ar_*), 152.2 (*C-12* or *C-14*), 153.8 (*C-12* or *C-14*), 172.3 (*C-10*, *C-16* or *C=O*), 173.6 (*C-10*, *C-16* or *C=O*), 174.6 (*C-10*, *C-16* or *C=O*). ESI-HRMS [M]^+^ m/z = 565.3654 (calculated for C_35_H_45_N_6_O: 565.3649).

#### 3.2.12. Compound **17**

To a solution of **16** (27 mg, 0.040 mmol, 1.0 eq.) in ACN (160 µL), 36 mg of **14** (0.060 mmol, 1.5 eq.), 40 µL of sodium ascorbate (1M in water, 0.04 mmol, 1 eq.), and 120 µL of copper (II) acetate (0.3 M in water, 0.036 mmol, 0.9 eq.) were added, and the mixture was stirred at room temperature for 16 h. The organic solvent was removed under reduced pressure, the aqueous layer was extracted with CH_2_Cl_2_ (3 × 20 mL) and dried over MgSO_4_, and the solvent was removed under vacuum. The crude product was purified using a flash chromatography on silica gel (eluent: DCM/MeOH 100/0 to 90/10) to afford compound **17**. Yield: 71% as a blue solid, *R_f_ =* 0.24 (DCM/MeOH: 95/5), Mp: 119 °C. IR (cm^−1^): ν = 3341, 2924, 2864, 1653, 1481, 1452, 1369, 1335, 1219. ^1^H NMR (CDCl_3_, 400 MHz): δ (ppm) = 1.06 (s, 21H, *Si-isopropyl*), 1.06 (s, 21H, *Si-isopropyl*), 1.53–1.64 (m, 2H, *H-20*), 1.68 (s, 12H, 4 *CH_3_*), 1.71–1.98 (m, 6H, *H-2, H-19* and *H-21)*, 2.19 (app qt, 2H, *J*_16,15_ = *J*_16,17_ = 6.5 Hz, *H-16*), 2.51 (br t, 2H, *J*_18,19_ = 7.0 Hz, *H-18*), 3.30 (br s, 2H, *H-17*), 3.57 (s, 3H, *N-CH_3_*), 3.63 (br t, 2H, *J*_2,1_ = 6.5 Hz, *H-1*), 4.01–4.12 (m, 1H, *H-6*), 4.15 (br t, 2H, *J*_22,21_ = 7.0 Hz, *H-22*), 4.16–4.22 (m, 1H, *H-3*), 4.24–4.33 (m, 2H, *H-4, H-5*), 4.35 (s, 2H, *H-8* or *H-11*), 4.37–4.51 (m, 2H, *H-7*), 4.43 (s, 2H, *H-8* or *H-11*), 4.52–4.61 (m, 4H, *H-14* and *H-15*), 6.25 (d, 1H, *J* = 14.0 Hz, *H-25* or *H-29*), 6.70 (d, 1H, *H-25* or *H-29*), 6.84 (app br t, 1H, *J*_27,26_ = *J*_27,28_ = 13.0 Hz, *H-27*), 7.07 (d, 1H, *J* = 7.5 Hz, *H_Ar_*), 7.14 (d, 1H, *J* = 7.5 Hz, *H_Ar_*), 7.18–7.26 (m, 2H, *H_Ar_*), 7.31–7.42 (m, 4H, *H_Ar_*), 7.77 (app t, 1H, *H-26* or *H-28*), 7.81 (app t, 1H, *H-26* or *H-28*), 8.20 (br s, 1H, *H-triazole*), 9.10 (br s, 1H, *NH*). ^13^C NMR (CDCl_3_, 100.6 MHz): δ (ppm) = 11.3 (3C, Si-*C*H(CH_3_)_3_), 11.3 (3C, *C*(CH_3_)_3_), 18.7 (6C, C(*C*H_3_)_3_), 18.7 (3C, Si-CH(*C*H_3_)_3_), 25.2 (*C-19*), 26.3 (*C-20*), 27.0 (*C-21*), 28.2 (2 *CH_3_)*, 28.3 (2 *CH_3_)*, 30.2 (*C-2*), 30.6 (*C-16*), 31.5 (N-*CH_3_*), 35.9 (*C-18*), 36.3 (*C-17*), 48.3 (*C-22*), 49.0 (*C-15*), 58.6 (*C-8* or *C-11*), 59.2 (*C-8* or *C-11*), 64.4 (*C-14*), 67.4 (*C-1*), 75.9 (*C-4*), 77.8 (*C-3*), 78.0 (d, *J_C5-F_* = 5 Hz, *C-5*), 78.6 (d, *J_C6-F_* = 18.5 Hz, *C-6*), 83.0 (d, *J_C7-F_* = 173.0 Hz, *C-7*), 88.6 (*C-10* or *C-13*), 88.8 (*C-10* or *C-13*), 102.5 (*C-9* or *C-12*), 103.2 (*C-9* or *C-12*), 103.6 (*C-25* or *C-29*), 105.4 (*C-25* or *C-29*), 110.3 (*C_Ar_*), 111.3 (*C_Ar_*), 122.2 (*C_Ar_*), 122.2 (*C_Ar_*), 124.3 (*CH-triazole*), 125.0 (*C_Ar_*), 125.7 (*C_Ar_*), 126.9 (*C-27*), 128.9 (*C_Ar_*), 129.1 (*C_Ar_*), 140.7 (*Cq_Ar_*), 141.3 (*Cq_Ar_*), 142.0 (*Cq_Ar_*), 143.0 (*Cq_Ar_*), 144.6 (*Cq-triazole*), 152.4 (*C-26* or *C-28*), 153.6 (*C-26* or *C-28*), 172.4 (*C-30* or *C-24*), 174.7(*C-30* or *C-24*). ^19^F NMR (CDCl3, 376 MHz): δ (ppm) = −230.7. ESI-HRMS [M + H]^2+^ m/z = 586.3942 (calculated for C_69_H_105_FN_6_O_5_Si_2_: 586.3829).

#### 3.2.13. Compound **18**

Prepared from **15** and **16** following the procedure described for **17**. Yield: 95% as a blue solid, *R_f_ =* 0.43 (DCM/MeOH: 95/5). IR (cm^−1^): ν = 3660, 2941, 2929, 2862, 2169, 2096, 1654, 1450, 1369, 1332, 1089.^1^H NMR (CDCl_3_, 400 MHz): δ (ppm) = 1.03 (s, 21H, *Si-isopropyl*), 1.04 (s, 9H, *Si-tert-butyl*), 1.05 (s, 21H, *Si-isopropyl*), 1.53–1.63 (m, 2H, *H-20*), 1.68 (s, 12H, 4 *CH_3_*), 1.79–1.99 (m, 6H, *H-2, H-19* and *H-21)*, 2.18 (app qt, 2H, *J*_16,15_ = *J*_16,17_ = 6.0 Hz, *H-16*), 2.56 (t, 2H, *J*_18,19_ = 7.0 Hz, *H-18*), 3.27–3.33 (br t, 2H, *H-17*), 3.55 (s, 3H, *N-CH_3_*), 3.57–3.76 (m, 4H, *H-1* and *H-7*), 4.00–4.04 (m, 1H, *H-6*), 4.10–4.18 (m, 3H, *H-22* and *H-3*), 4.22–4.29 (m, 2H, *H-4, H-5*), 4.26 (d, 1H, *J* = 16.0 Hz, Hz, *H-8a* or *H-11a)*, 4.32 (d, 1H, *J* = 16.0 Hz, Hz, *H-8b* or *H-11b)*, 4.43 (s, 2H, *H-8* or *H-11*), 4.53 (bt, 1H, *H-15*), 4.55 (d, 1H, *J* = 12.0 Hz, *H-14a*), 4.59 (d, 1H, *H-14b*), 6.23 (d, 1H, *J* = 13.5 Hz, *H-25* or *H-29*), 6.67 (d, 1H, *H-25* or *H-29*), 6.97 (br t, 1H, *J*_27,26_ = *J*_27,28_ = 12.0 Hz, *H-27*), 7.05 (d, 1H, *J* = 7.5 Hz, *H_Ar_*), 7.13 (d, 1H, *J* = 7.5 Hz, *H_Ar_*), 7.17–7.25 (m, 2H, *H_Ar_*), 7.30–7.42 (m, 10H, *H_Ar_*), 7.64–7.70 (m, 4H, *H_Ar_*), 7.78 (app t, 1H, *H-26* or *H-28*), 7.82 (app t, 1H, *H-26* or *H-28*), 8.07 (s, 1H, *H-triazole*), 9.37 (br s, 1H, *NH*). ^13^C NMR (CDCl_3_, 100.6 MHz): δ (ppm) = 11.3 (3C, Si-*C*H-(CH_3_)_2_), 11.3 (3C, Si-*C*H-(CH_3_)_2_), 18.7 (6C, Si-CH-(*C*H_3_)_2_), 18.7 (6C, Si-CH-(*C*H_3_)_2_), 19.4 (C, Si-*C*), 25.3 (*C-19*), 26.3 (*C-20*), 26.9 (*C-21*), 27.0 (3C, Si-C*(CH_3_)_3_*), 28.2 (2 *CH_3_)*, 28.3 (2 *CH_3_)*, 30.3 (*C-2*), 30.4 (*C-16*), 31.5 (N-*CH_3_*), 35.6 (*C-18*), 36.4 (*C-17*), 44.8 (*C-22*), 48.4 (*C-15*), 49.0 (*C-23* or *C-31*), 49.5 (*C-23* or *C-31*), 58.5 (*C-8* or *C-11*), 59.1 (*C-8* or *C-11*), 64.2 (*C-14*), 64.8 (*C-7*), 67.9 (*C-1*), 76.8 (*C-3* and *C-4*), 79.3 (*C-5*), 80.2 (*C-6*), 87.9 (*C-10* or *C-13*), 88.0 (*C-10* or *C-13*), 103.3 (*C-9* or *C-12*), 103.5 (*C-9* or *C-12*), 103.6 (*C-25* or *C-29*), 105.3 (*C-25* or *C-29*), 110.3 (*C_Ar_*), 111.3 (*C_Ar_*), 122.2 (*C_Ar_*), 122.3 (*C_Ar_*), 124.2 (*CH-triazole*), 125.0 (*C_Ar_*), 125.7 (*C_Ar_*), 126.8 (*C-27*), 127.8 (2*C_Ar_*), 127.8 (2*C_Ar_*), 128.8 (*C_Ar_*), 129.1 (*C_Ar_*), 129.8 (2*C_Ar_*), 133.5 (*Cq_Ar_*), 133.6 (*Cq_Ar_*), 135.7 (2*C_Ar_*), 135.8 (2*C_Ar_*), 140.7 (*Cq_Ar_*), 141.3 (*Cq_Ar_*), 142.0 (2*Cq_Ar_*), 142.9 (*Cq_Ar_*), 144.6 (*Cq-triazole*), 152.4 (*C-26* or *C-28*), 153.6 (*C-26* or *C-28*), 172.5 (*C-30, C-24* or *C=O*), 173.6 (*C-30, C-24* or *C=O*), 174.9 (*C-30, C-24* or *C=O*). ESI-HRMS [M + H]^2+^ m/z = 704.9436 (calculated for C_85_H_125_N_6_O_6_Si_3_: 704.9470).

#### 3.2.14. Compound **19**

To a solution of **17** (30 mg, 0.023 mmol, 1.0 eq.) in THF (1 mL), 54 µL of tetrabutylammonium fluoride (1 M in THF, 0.054 mmol, 2.4 eq.) was added at 0 °C under an inert atmosphere, and the mixture was stirred at 0 °C for 3 h. The organic solvent was removed under reduced pressure, and the residue was solubilized in CH_2_Cl_2_ (10 mL). The organic layer was washed with 0.01M HCl solution (2 × 5 mL) and with brine until pH = 7. The organic layer was dried over MgSO_4_, and the solvent was removed under vacuum. The crude product was purified using a flash chromatography on silica gel (eluent: DCM/MeOH 100/0 to 90/10) to afford compound **19**. Yield: 68% as a blue solid, *R_f_ =* 0.05 (DCM/MeOH: 95/5), Mp: 136 °C. IR (cm^−1^): ν = 2970, 2912, 1647, 1477, 1448, 1367, 1331, 1217. ^1^H NMR (CDCl_3_, 400 MHz): δ (ppm) = 1.50–1.61 (m, 2H, *H-20*), 1.68 (s, 12H, 4 *CH_3_*), 1.71–2.00 (m, 6H, *H-2, H-19* and *H-21)*, 2.15–2.21 (m, 2H, *H-16*), 2.48 (br t, 2H, *J*_18,19_ = 7.5 Hz, *H-18*), 2.48–2.52 (m, 2H, *H-10* and *H-13*), 3.24–3.31 (m, 2H, *H-17*), 3.56 (s, 3H, *N-CH_3_*), 3.57–3.63 (m, 2H, *H-1*), 3.96–4.08 (m, 1H, *H-6*), 4.12 (br t, 2H, *J*_22,21_ = 7.0 Hz, *H-22*), 4.15–4.20 (m, 3H, *H-3, H-4* and *H-5*), 4.27 (d, 2H, *J* = 2.5 Hz, *H-8* or *H-11*), 4.37 (d, 2H, *J* = 2.5 Hz, *H-8* or *H-11*), 4.37–4.51 (m, 2H, *H-7*), 4.51–4.63 (m, 4H, *H-14* and *H-15*), 6.22 (d, 1H, *J* = 13.5 Hz, *H-25* or *H-29*), 6.60 (d, 1H, *H-25* or *H-29*), 6.91 (app br t, 1H, *J*_27,26_ = *J*_27,28_ = 12.0 Hz, *H-27*), 7.07 (d, 1H, *J* = 8.0 Hz, *H_Ar_*), 7.14 (d, 1H, *J* = 8.0 Hz, *H_Ar_*), 7.19–7.26 (m, 2H, *H_Ar_*), 7.32–7.41 (m, 4H, *H_Ar_*), 7.77 (app t, 1H, *H-26* or *H-28*), 7.81 (app t, 1H, *H-26* or *H-28*), 8.22 (s, 1H, *H-triazole*), 9.00 (br s, 1H, *NH*). ^13^C NMR (CDCl_3_, 100.6 MHz): δ (ppm) = 25.3 (*C-19*), 26.4 (*C-20*), 27.0 (*C-21*), 28.2 (2 *CH_3_)*, 28.3 (2 *CH_3_)*, 29.7 (*C-2*), 30.4 (*C-16*), 31.9 (N-*CH_3_*), 35.8 (*C-18*), 36.2 (*C-17*), 44.9 (*C-22*), 48.4 (*C-15*), 58.4 (*C-8* or *C-11*), 58.5 (*C-8* or *C-11*), 64.1 (*C-14*), 67.2 (*C-1*), 75.1 (*C-10* or *C-13*), 75.5 (*C-10* or *C-13*), 76.4 (*C-4*), 77.9 (*C-3*), 78.0 (d, *J_C6-F_* = 18.5 Hz, *C-6*), 79.2 (d, *J_C5-F_* = 5 Hz, *C-5*), 79.3 (*C-9* or *C-12*), 79.9 (*C-9* or *C-12*), 82.7 (d, *J_C7-F_* = 172.0 Hz, *C-7*), 103.8 (*C-25* or *C-29*), 105.4 (*C-25* or *C-29*), 110.2 (*C_Ar_*), 111.2 (*C_Ar_*), 122.1 (*C_Ar_*), 122.1 (*C_Ar_*), 124.9 (*CH-triazole*), 125.0 (*C_Ar_*), 125.6 (*C_Ar_*), 126.4 (C-27), 128.7 (*C_Ar_*), 129.0 (*C_Ar_*), 140.6 (*Cq_Ar_*), 141.1 (*Cq_Ar_*), 141.8 (*Cq_Ar_*), 142.8 (*Cq_Ar_*), 144.1 (*Cq-triazole*), 152.3 (*C-26* or *C-28*), 153.5 (*C-26* or *C-28*), 172.3 (*C-30* or *C-24*), 174.7 (*C-30* or *C-24*). ^19^F NMR (CDCl_3_, 376 MHz): δ (ppm) = −231.1. ESI-HRMS [M + H]^2+^ m/z = 430.2592 (calculated for C_51_H_64_FN_6_O_5_: 430.2495). Analytical HPLC analyses were performed using a waters system. Condition du gradient Rt = 4. 4 min, ACN/H_2_O 60/40 (*v*/*v*) with 0.1% TFA in isocratic conditions, flow rate of 1.5 mL/min, and UV detection (650 nm).

#### 3.2.15. Compound **20**

Prepared from **18** following the procedure described for **19**. Yield: 69% as a blue solid, *R_f_ =* 0.15 (DCM/MeOH: 9/1). IR (cm^−1^): ν = 3236, 2926, 2162, 1651, 1495, 1495, 1454, 1371, 1091. ^1^H NMR (CDCl_3_, 400 MHz): δ (ppm) = 1.57 (app qt, 2H, *J*_20,19_ = *J*_20,21_ = 6.5 Hz, *H-20*), 1.68 (s, 12H, 4 *CH_3_*), 1.78–1.94 (m, 6H, *H-2, H-19* and *H-21)*, 2.19 (app qt, 2H, *J*_16,15_ = *J*_16,17_ = 6.5 Hz, *H-16*), 2.44 (t, 1H, *J* = 2.0 Hz, *H-10 or H-13*), 2.47 (t, 1H, *J* = 2.0 Hz, *H-10 or H-13*), 2.50 (br t, 2H, *J*_18,19_ = 7.0 Hz, *H-18*), 2.59 (br s, 1H, OH), 3.25–3.32 (m, 2H, *H-17*), 3.56 (s, 3H, *N-CH_3_*), 3.57–3.66 (m, 2H, *H-1*), 3.66 (dd, 1H, *J*_7a,7b_ = 12.0 Hz, *J*_7a,6_ = 3.5 Hz, *H-7a*), 3.83 (dd, 1H, *J*_7a,7b_ = 12.0 Hz, *J*_7b,6_ = 2.5 Hz, *H-7b*), 3.95 (app dt, 1H, *J*_6,5_ = 7.5 Hz, *H-6*), 4.13 (br t, 2H, *J*_22,21_ = 7.0 Hz, *H-22*), 4.16 (br t, 1H, *J*_4,3_ = *J*_4,5_ = 4.0 Hz, *H-4*), 4.22–4.28 (m, 2H, *H-3, H-5*), 4.30 (d, 2H, *H-8* or *H-11*), 4.38 (d, 2H, *H-8* or *H-11*), 4.57 (br t, 2H, *J*_15,16_ = 6.5 Hz, *H-15*), 4.58 (d, 1H, *J*_14a,14b_ = 12.5 Hz, *H-14a*), 4.68 (d, 1H, *H-14b*), 6.21 (d, 1H, *J* = 13.5 Hz, *H-25* or *H-29*), 6.61 (d, 1H, *H-25* or *H-29*), 6.91 (app br t, 1H, *J*_27,26_ = *J*_27,28_ = 12.5 Hz, *H-27*), 7.07 (d, 1H, *J* = 8.0 Hz, *H_Ar_*), 7.15 (d, 1H, *J* = 8.0 Hz, *H_Ar_*), 7.19–7.28 (m, 2H, *H_Ar_*), 7.31–7.42 (m, 4H, *H_Ar_*), 7.75 (app t, 1H, *H-26* or *H-28*), 7.79 (app t, 1H, *H-26* or *H-28*), 8.37 (s, 1H, *H-triazole*), 9.09 (br s, 1H, *NH*). ^13^C NMR (CDCl_3_, 100.6 MHz): δ (ppm) = 25.1 (*C-19*), 26.2 (*C-20*), 26.9 (*C-21*), 28.1 (2 *CH_3_)*, 28.1 (2 *CH_3_)*, 29.9 (*C-2*), 30.3 (*C-16*), 31.3 (N-*CH_3_*), 35.8 (*C-18*), 36.0 (*C-17*), 44.7 (*C-22*), 48.2 (*C-15*), 48.9 (*C-23* or *C-31*), 49.4 (*C-23* or *C-31*), 58.4 (*C-8* or *C-11*), 58.5 (*C-8* or *C-11*), 61.9 (*C-7*), 64.3 (*C-14*), 66.8 (*C-1*), 74.6 (*C-10* or *C-13*), 75.1 (*C-10* or *C-13*), 77.2 (*C-3*), 77.3 (*C-4*), 79.9 (*C-9* or *C-12*), 80.0 (*C-5*), 80.1 (*C-6*), 80.2 (*C-9* or *C-12*), 103.3 (*C-25* or *C-29*), 105.0 (*C-25* or *C-29*), 110.2 (*C_Ar_*), 111.2 (*C_Ar_*), 122.1 (*C_Ar_*), 122.1 (*C_Ar_*), 124.6 (*CH-triazole*), 125.0 (*C_Ar_*), 125.7 (*C_Ar_*), 126.4 (C-27), 128.7 (*C_Ar_*), 129.0 (*C_Ar_*), 140.5 (*Cq_Ar_*), 141.1 (*Cq_Ar_*), 141.8 (*Cq_Ar_*), 142.8 (*Cq_Ar_*), 142.5 (*Cq_Ar_*), 144.5 (*Cq-triazole*), 152.2 (*C-26* or *C-28*), 153.4 (*C-26* or *C-28*), 172.5 (*C-30, C-24* or *C=O*), 173.5 (*C-30, C-24* or *C=O*), 174.6 (*C-30, C-24* or *C=O*). ESI-HRMS [M+]^2+^ m/z = 429.2568 (calculated for C_51_H_65_N_6_O_6_: 429.2516).

#### 3.2.16. Compound **22**

To a solution of c(RGDfK) (10 mg, 14.0 µmol, 1.0 eq.) in DMF (1 mL), 11 mg of azido-PEG_4_-NHS (28.0 µmol, 2.0 eq.) and 5 µL of Et_3_N (35.0 µmol, 2.5 eq.) were added, and the mixture was stirred at 30 °C for 16 h. The organic solvent was evaporated under vacuum and the solid residue was washed with diethyl ether. The obtained solid was dried under vacuum to afford compound **22**. Yield: 72% as white powder. ^1^H NMR (D_2_O, 400 MHz): δ (ppm) = 0.84–0.96 (m, 2H), 1.28–1.40 (m, 3H), 1.44–1.60 (m, 3H), 1.62–1.76 (m, 2H), 1.85–1.94 (m, 1H), 2.53–2.58 (m, 3H), 2.71 (dd, 1H, *J* = 6.8 Hz, *J* = 15.8 Hz), 2.93 (dd, 1H, *J* = 11.0 Hz, *J* = 12.5 Hz), 3.09–3.26 (m, 4H), 3.48–3.53 (m, 2H), 3.70–3.75 (m, 14H), 3.81 (t, 2H, *J* = 6.0 Hz), 3.81 (app t, 1H, *J* = 6.0 Hz), 3.87 (dd, 1H, *J* = 10.5 Hz, *J* = 4.0 Hz), 4.24 (dd, 1H, *J* = 15.0 Hz), 4.43 (dd, 1H, *J* = 8.5 Hz, *J* = 5.7 Hz), 4.60 (dd, 1H, *J* = 10.7 Hz, *J* = 5.7 Hz), 4.73 (app t, 1H, *J* = 7.1 Hz), 7.27–7.36 (m, 3H), 7.38–7.42 (m, 2H). HRMS [M]^+^ m/z = 877.4522 (calculated for C_38_H_61_N_12_O_12_: 877.4526).

#### 3.2.17. Compound **23**

To a solution of **19** (3 mg, 3.04 µmol, 1.0 eq.) in a mixture of water/ACN (3/2.5) (550 µL), 7.8 mg of **21** (9.12 µmol, 3.0 eq.), 9 µL of copper (II) sulphate (1 M in water, 9.12 µmol, 3.0 eq.), and 23 µL of sodium ascorbate (1 M in water, 22.80 µmol, 7.5 eq.) were added, and the mixture was stirred at 40 °C for 24 h. Chelex^®^ 100 resin (100 mg) was then added to the solution, and the suspension was stirred for 10 min. The resin was filtered off and the resulting solution dried under vacuum. The crude product was purified using Sephadex LH20 in water/ACN (7/3) to afford compound **23**. Yield: 42% as a blue solid. HRMS [M + H]^4+^ m/z = 586.8230 (calculated for C_117_H_167_FN_30_O_21_: 586.8221), [M]^3+^ m/z = 782.0925 (calculated for C_117_H_166_FN_30_O_21_: 782.0937).

#### 3.2.18. Compound **24**

Prepared from **19** and **22** following the procedure described for **23**. Yield: 62% as a blue solid. HRMS [M + H]^4+^ m/z = 653.8509 (calculated for C_127_H_187_FN_30_O_29_: 653.8511), [M]^3+^ m/z = 871.4603 (calculated for C_127_H_186_FN_30_O_29_: 871.4657).

#### 3.2.19. Compound **25**

To a solution of c(RGDfK) (9 mg, 12.5 µmol, 1.0 eq.) in DMF (1 mL), 8.8 mg of commercial NHS cyanine 5 (12.5 µmol, 2.0 eq.) and 5 µL of Et_3_N (50.0 µmol, 4.0 eq.) were added, and the mixture was stirred at 50 °C for 16 h. The organic solvent was evaporated under vacuum, and the crude product was purified using a semi-preparative HPLC with a C18 reversed-phase silica gel: solvent A: 0.1% TFA water; solvent B: ACN; 0 to 2 min: 5% to 20% B, 2 to 5 min, 20% to 30% B, 5 to 20 min, 30% to 100%, 20 to 22 min, 100% to 5% B. Flow rate: 10 mL/min. The resulting solution was freeze-dried to afford compound **25**. Yield: 59% as a blue solid, *R_f_* = 0.03 (DCM/MeOH: 8/2), *T_R_* = 22.0 min. ^1^H NMR (CDCl_3_, 400 MHz): δ (ppm) = 0.79–0.93 (m, 2H), 1.17–1.55 (m, 9H), 1.61 (s, 12H), 1.62–1.72 (m, 2H), 1.78–1.93 (m, 3H), 2.25 (t, 2H, *J* = 7.0 Hz), 2.64–2.72 (m, 1H), 2.75–2.89 (m, 2H), 2.93–3.06 (m, 3H), 3.09–3.20 (m, 2H), 3.51 (d, 1H, J = 14.5 Hz), 3.59 (s, 3H), 3.74–3.84 (m, 1H), 4.10 (bt, 2H, *J* = 7.5 Hz), 4.20 (bt, 1H), 4.31–4.71 (m, 3H), 6.13–6.21 (m, 2H), 6.44 (app br t, 1H, *J* = 12 Hz), 7.16–7.33 (m, 8H), 7.37–7.46 (m, 3H), 7.48–7.55 (m, 2H), 7.89–8.00 (m, 2H). ESI-HRMS [M]^2+^ m/z = 534.8034 (calculated for C_59_H_79_N_11_O_8_: 534.8051).

#### 3.2.20. Compound **26**

A solution of **20** (9 mg, 9.14 µmol) in ACN (300 µL) was diluted in water (60 mL). The obtained solution was passed through a series of 4 Oasis^®^ MCX cartridges to trap the compound. The cartridges were washed with 100 mL of water (until pH = 7). The product was eluted with a mixture of NaOTf (0.2 M)/ACN (1/9) (100 mL), and the solvent was evaporated under vacuum. The crude product was solubilized in CH_2_Cl_2_ (10 mL), washed with water (2 × 5 mL), and dried over MgSO_4_, and the solvent was removed under vacuum. Compound **26** was obtained quantitatively without further purification. The quantitative yield is a blue solid. ^1^H NMR (CDCl_3_, 400 MHz): δ (ppm) = 1.47–1.58 (m, 2H, *H-20*), 1.67 (s, 6H, 4 *CH_3_*), 1.68 (s, 6H, 4 *CH_3_*), 1.70–1.94 (m, 6H, *H-2*, *H-19* and *H-21*), 2.06–2.15 (m, 2H, *H-16*), 2.33 (br t, 2H, *J*_18,19_ = 6.5 Hz, *H-18*), 2.43 (t, 1H, *J* = 2.0 Hz, *H-10* or *H-13*), 2.48 (t, 1H, *J* = 2.0 Hz, *H-10* or *H-13*), 3.19–3.26 (m, 2H, *H-17*), 3.56 (s, 3H, *N-CH_3_*), 3.57–3.66 (m, 3H, *H-1*, *H-7a*), 3.80 (dd, 1H, *J*_7a,7b_ = 12.0 Hz, *J*_7b,6_ = 2.5 Hz, *H-7b*), 3.93 (app dt, 1H, *J*_6,5_ = 7.5 Hz, *H-6*), 4.01 (br t, 2H, *J*_22,21_ = 7.0 Hz, *H-22*), 4.15 (br t, 1H, *J*_4,3_ = *J*_4,5_ = 3.5 Hz, *H-4*), 4.17–4.24 (m, 2H, *H-3*, *H-5*), 4.26 (d, 2H, *H-8* or *H-11*), 4.37 (d, 2H, *H-8* or *H-11*), 4.40 (br t, 2H, *J*_15,16_ = 6.0 Hz, *H-15*), 4.55 (d, 1H, *J*_14a,14b_ = 12.5 Hz, *H-14a*), 4.61 (d, 1H, *H-14b*), 6.24 (d, 1H, *J* = 13.5 Hz, *H-25* or *H-29*), 6.29 (d, 1H, *H-25* or *H-29*), 6.78 (app br t, 1H, *J*_27,26_ = *J*_27,28_ = 12.5 Hz, *H-27*), 7.09 (d, 1H, *J* = 8.0 Hz, *H_Ar_*), 7.11 (d, 1H, *J* = 8.0 Hz, *H_Ar_*), 7.19–7.27 (m, 2H, *H_Ar_*), 7.31–7.41 (m, 4H, *H_Ar_*), 7.44 (br s, 1H, *NH*), 7.80 (app t, 2H, *H-26* and *H-28*), 7.93 (s, 1H, *H-triazole*). ^19^F NMR (CD_3_CN, 376 MHz: δ −79.27.

#### 3.2.21. Compound **27**

To a solution of **26** (1.5 mg, 1.52 µmol, 1.0 eq.) in CH_2_Cl_2_ (1 mL), 1.33 μL of DIPEA (7.61 µmol l, 5.0 eq.) and 1.06 mg of methanesulfonic anhydride (6.09 µmol, 4.0 eq.) were added under an inert atmosphere. The mixture was stirred for 16 h at room temperature. The solution was evaporated under vacuum. Yield: 38% as a blue solid. ^1^H NMR (400 MHz, CD_3_CN) δ (ppm) = 1.39–1.45 (m, 2H, *H-20*), 1.64 (app qt, 2H, *J*_19,18_ = *J*_19,20_ = 6.5 Hz, *H-19*), 1.68 (s, 12H, 4 *CH_3_*), 1.73–1.88 (m, 4H, *H-21* and *H-2)*, 2.00 (app qt, 2H, *J*_16,15_ = *J*_16,17_ = 6.5 Hz, *H-16*), 2.12–2.17 (m, 2H, *H-18*), 2.75 (t, 1H, *J* = 2.0 Hz, *H-10* or *H-13*), 2.79 (t, 1H, *J* = 2.0 Hz, *H-10* or *H-13*), 3.04 (s, 3H, *CH_3_-Ms*), 3.09–3.14 (m, 2H, *H-17*), 3.54 (s, 5H, *H-1*, *N-CH_3_*), 3.96 (ddd, 1H, *J*_6,5_ = 7.5 Hz, *J*_6,7a_ = 4.5 Hz, *J*_6,7b_ = 2.5 Hz, *H-6*), 4.02 (br t, 2H, *J*_22,21_ = 7.0 Hz, *H-22*), 4.06–4.12 (m, 3H, *H-3, H-5, H-4*), 4.21 (dd, 1H, *J*_7a,7b_ = 11.5 Hz, *J*_7a,6_ = 4.5 Hz, *H-7a*), 4.24–4.27 (m, 2H, *H-8* or *H-11*), 4.30–4.39 (m, 5H, H-7b, H-15, *H-8* or *H-11*), 4.51 (s, 2H, *H-14*), 6.19 (d, 1H, *J* = 13.5 Hz, *H-25* or *H-29*), 6.25 (d, 1H, *H-25* or *H-29*), 6.55 (app br t, 1H, *J*_27,26_ = *J*_27,28_ = 12.5 Hz, *H-27*), 6.93 (t, 1H, *J* = 6.0 Hz, *NH*), 7.21–7.28 (m, 4H, *H_Ar_*), 7.33–7.43 (m, 2H, *H_Ar_*), 7.45–7.50 (m, 2H, *H_Ar_*), 7.88 (s, 1H, *H-triazole*), 8.08 (app t, 2H, *H-26* and *H-28*). ^13^C NMR (CD_3_CN, 100.6 MHz): δ (ppm) = 26.0 (*C-19*), 27.0 (*C-20*), 27.6 (2 *CH_3_)*, 27.8 (3C, 2 *CH_3_* and *C-21)*, 30.4 (*C-2*), 31.1 (*C-16*), 32.0 (N-*CH_3_*), 36.5 (*C-18*), 36.9 (*C-17*), 37.7 (CH_3_-Ms), 44.9 (*C-22*), 48.5 (*C-15*), 50.1 (*C-23* or *C-31*), 50.2 (*C-23* or *C-31*), 59.0 (*C-8* or *C-11*), 59.4 (*C-8* or *C-11*), 64.8 (*C-14*), 67.6 (*C-1*), 71.1 (*C-7*), 76.0 (*C-10* or *C-13*), 76.6 (*C-10* or *C-13*), 77.8 (*C-6*), 78.1 (*C-4*), 78.6 (*C-3*), 80.6 (*C-9* or *C-12*), 80.9 (*C-5*), 81.0 (*C-9* or *C-12*), 104.1 (2C, *C-25* and *C-29*), 111.8 (*C_Ar_*), 112.1 (*C_Ar_*), 123.1 (*C_Ar_*), 123.2(*C_Ar_*), 124.6 (*CH-triazole*), 125.6 (*C_Ar_*), 125.9 (*C_Ar_*), 126.0 (*C-27*), 129.5 (*C_Ar_*), 129.5 (*C_Ar_*), 142.3 (*Cq_Ar_*), 142.4 (*Cq_Ar_*), 143.4 (*Cq_Ar_*), 144.1 (*Cq_Ar_*), 145.6 (*Cq_Ar_*), 144.7 (*Cq-triazole*), 154.8 (*C-26* or *C-28*), 154.9 (*C-26* or *C-28*), 173.7 (*C-30, C-24* or *C=O*), 174.4 (*C-30, C-24* or *C=O*), 174.9 (*C-30, C-24* or *C=O*). ^19^F NMR (376 MHz, CD_3_CN) δ (ppm) = -79.31. ESI-HRMS [M+]^+^ m/z = 935.4562 and [M+]^2+^ m/z = 468.2452 (calculated for C_52_H_67_N_6_O_8_S, respectively: 935.4736 and 468.2404).

### 3.3. ^18^F-Radiolabeling of the Precursor 27

#### 3.3.1. Protocol on AIO Synthesizer

[^18^F]Fluoride (~1000 MBq) in H_2_[^18^O]O was recovered in the AIO synthesizer and passed through a Sep-Pak^®^ light QMA-carbonate cartridge, where [^18^F]fluoride was trapped and H_2_[^18^O]O was collected for recycling. The QMA-carbonate cartridge was then flushed with nitrogen gas flow. The trapped [^18^F]fluoride was eluted from the QMA-carbonate cartridge into the reactor with 1mL of the K_222_/K_2_CO_3_ solution (K_222_/K_2_CO_3_ 15 mg/1.3 mg in ACN/H_2_O 8/2 v/v). The solvent was removed under a stream of nitrogen gas flow at 110 °C for 10 min to give the dried K[^18^F]F-K_222_ complex. The reactor was then cooled at 75 °C to perform the ^18^F-radiofluorination. The mesylated precursor **27** (5.3 mg) diluted in 2 mL of ACN was transferred into the reactor containing the dried K[^18^F]F-K_222_ complex. Radiolabeling was performed at 95 °C for 10 min in a closed reactor thanks to pinch valves. After cooling at 30 °C, the reaction mixture was pushed into the final product vial. **[^18^F]19** was produced with a decay-corrected radiochemical yield of 13%, as determined using radio-HPLC analyses.

#### 3.3.2. Protocol on TracerLab FxFN Synthesizer

[^18^F]Fluoride (~17.0 GBq) in H_2_[^18^O]O was recovered in the TracerLab FxFN synthesizer and passed through a Sep-Pak^®^ light QMA-carbonate cartridge, where [^18^F]fluoride was trapped and H_2_[^18^O]O was collected for recycling. The trapped [^18^F]fluoride was eluted from the QMA-carbonate cartridge into the reactor with 1 mL of the K_222_/K_2_CO_3_ solution (K_222_/K_2_CO_3_ 12 mg/2 mg in ACN/H_2_O 7/3 *v*/*v*). The solvent was removed in two heating steps: first, at 60 °C for 7 min at a pressure ranging between 30 and 35 kPa and, then, at 120 °C for 5 min under vacuum to give the dried K[^18^F]F-K_222_ complex. The reactor was then cooled at 50 °C to perform the ^18^F-radiofluorination. The mesylated precursor **27** (3.8 mg) diluted in 0.7 mL of ACN was transferred into the reactor containing the dried K[^18^F]F-K_222_ complex. Radiolabeling was performed at 95 °C for 10 min in a closed reactor. After cooling at 40 °C, the reaction mixture was pushed into the final product vial. **[^18^F]19** was produced with a decay-corrected radiochemical yield of 11%, as determined using radio-HPLC analyses.

For both protocols, an aliquot of **[^18^F]19** was injected on the analytical HPLC (20 µL) and a second injection with the corresponding non-radioactive compound **19** confirmed the identity of **[^18^F]19**. A Luna PFP column was used in isocratic conditions; eluent: can/H_2_O 60/40 v/v with 0.1% of TFA, flow: 1.0 mL/min or 1.5 mL/min, and UV detection (650 nm).

### 3.4. Absorption and Fluorescence Measurements

Absorption spectra were recorded in diluted solution (µM) in an aqueous PBS buffer (0.01 M, pH 7.4). Fluorescence quantum yields *Φ_fluo_* were measured in diluted solutions with an absorbance lower than 0.1 using the following equation:$$\phi_{fluo,\;\;x}=\phi_{fluo,ref}\left(\frac{Grad_{x}}{Grad_{ref}}\right)\left(\frac{n_{x}^{2}}{n_{ref}^{2}}\right)$$
where *Φ_fluo_* is the fluorescence quantum yield; *Grad* is the gradient from the plot of integrated fluorescence intensity vs. absorbance; *n* the refractive index of the solvent; and the subscripts *x* and *ref* denote sample and reference. The fluorescence quantum yields of **23** and **24** were measured relative to the commercial Cy5.Cl in PBS for which *Φ_fluo,ref_* = 0.13 [36]. The excitation of the reference and sample compounds was performed at the same wavelength (λ_ex_ = 640 nm).

### 3.5. Integrin α_v_β_3_ Binding Assay

The affinity of the compounds for integrin protein was evaluated in terms of the half maximal inhibitory concentration (IC_50_ values) using a solid-phase assay, as previously described by Tobias G. Kapp et al. [52]. Briefly, the surface of Maxisorp microplates (NUNC, ThermoFicher Scientific, Paris, France) was coated with 1 µg/mL human vitronectin (Bio-techne, Lille, France) overnight at +4 °C. The non-specific sites were blocked with a TSB buffer (20 mM Tris-HCl, 150 mM NaCl, 1 mM CaCl_2_, 1 mM MgCl_2_, 1 mM MnCl_2_, pH 7.5, 1% BSA) (Sigma-Aldrich, Saint-Quentin-Fallavier, France) for 1 h at 37 °C. The binding of the compounds was assessed using 2 µg/mL integrin α_v_β_3_ (Bio-techne, Lille, France) in the presence of the serial dilutions of the compounds or the reference compound c(RGDfK) as positive control. After 1 h incubation at room temperature, the plates were washed, and the amount of bound integrin was stained by incubation with 2 µg/mL of mouse anti-human CD51/61 (BD Pharmingen, Paris, France) and 1 µg/mL of anti-mouse IgG horseradish peroxidase conjugate antibody (Bio-techne, France). The enzymatic reaction was carried out in the dark by the addition of the enzyme substrate (Bio-techne, France) and stopped after 10 min by the addition of H_2_SO_4_ (Stop Solution, Bio-techne, France). The optical densities were measured at 450 nm. The values were blank subtracted, and the results were expressed as relative absorbance percentage in comparison to the wells containing only integrin α_v_β_3_.

Affinities were estimated from 3 independent series performed in duplicate as IC_50_ values (i.e., the concentration of the compounds that displaced 50% of integrin binding calculated using one-site fit log-IC_50_ non-linear analysis regression using the GraphPad Prism 6 software (v 6.05, USA)).

### 3.6. Cellular Uptake

Human glioblastoma U-87 MG cells were cultured in DMEM (Dulbecco’s Modified Eagle Medium) and supplemented with sodium pyruvate (1.5 mM) and vitamins MEM AA, MEM NE AA, L-Ser (14 μg/mL), L-asp (25 μg/mL), L-Glu (2.5 mM), penicillin (100 U/mL), streptomycin (100 μg/mL) and 20% fetal calf serum (FCS) under standard cell culture conditions at 37 °C in a humidified atmosphere (80%) containing 5% CO_2_.

U-87 MG cells were seeded in 12 multi-well plates at 10 × 10^4^ cells/cm^2^ and cultivated for 48 h. The old culture medium was discarded, and the cells were exposed to 1–10 µM of compound for 1, 4, or 24 h. After 3 washes, the U-87MG cells were then detached from their support by trypsination, centrifuged for 10 min at 300 g, and suspended in 1 mL of HBSS. An aliquot of cell suspension was taken for numeration (TC20, Biorad. Hercules, CA, USA), and the rest of the cells were centrifuged. Afterwards, the pellet was resuspended in DMSO and sonicated in a water bath for 10 min to lyse the cells and solubilize the Cy5 conjugates. The fluorescence signals of Cy5 in the samples were measured in duplicate at an excitation wavelength of 645/9 nm and an emission wavelength of 680/20 nm (Tecan Infinite M200 Pro spectrofluorometer, Tecan, Männedorf, Switzerland). The fluorescence signals of known concentrations of Cy5-conjugates diluted in DMSO were used to draw a standard calibration curve. The concentration of Cy5 present in the samples (nM) was determined from the linear regression analysis of the standard calibration curve (Equation (1)). The cell count (number of cells per mL) was used to calculate the number of cells present in the samples (number of cell per 850 μL). The results of the cellular uptake were expressed as the concentration of Cy5 (nM) incorporated per one million of cells (Equation (2)).
[Cy5] (nM) = f (Fluorescence Intensity)(1)
[Cy5] (nM per million of cells) = ([Cy5] (nM) × 10^6^ cells)/Number of cells in the sample(2)

### 3.7. Confocal Fluorescence Microscopy

A total of 96 multi-well round bottom plates (Costar 3799) were coated with 50 µL of 25 mg/mL poly-hemma (Sigma-Aldrich, St. Louis, MO, USA). After evaporation at room temperature, the wells were seeded at 250 U-87 MG cells/well and incubated at 37 °C, 5% CO_2_, and 80% humidity. After 7 days, the spheroids (250 µm diameter) were exposed to 1–10 µM of compound for 1, 4, or 24 h. After several washes, the spheroids were fixed using paraformaldehyde 4% for 30 min. ImageXPress microconfocal (Molecular Devices, San Jose, CA, USA) was used to perform fluorescence microscopy imaging (50 µm slit spinning disk, exposure time: Cy5 200 msec (λ_ex_ 631/28 nm/λ_em_ 692/40 nm, dichroïque 660 nm). Z-series images were performed using 10× magnification (Plan Fluor NIKON, Tokyo, Japan) (step size 2 µm; number of steps 133), and 2D projection was realized using best focus algorithm (scale bar 100 µm). For tumor imaging, the ectopic tumors were harvested and cooled in a tissue freezing medium (MM France, Brignais, France) prior to slicing (10 µm sections) with cryostat. Z-series images were performed using 4× and 10× magnification using ImageXPress microconfocal (Molecular Devices, San Jose, CA, USA).

### 3.8. Animal Models and NIRF Imaging

Animal experiments were performed in accordance with the protocols approved by the French Ministry of Research after a review by the local animal protection and use committee (APAFIS# 30902). For ectopic tumor biodistribution, a U-87 MG tumor model was established by subcutaneous injection of U-87 MG cells (2 × 10^6^ in 100 µL of 5% glucose solution) into the front right flank of female athymic nude mice (Charles Rivers, Wilmington, MA, USA). The mice were subjected to imaging studies when the tumor volume reached 500 mm^3^ (3–4 wk after inoculation) at 1, 4, 6, and 24 h after the intravenous administration of the compounds **24** and **25** at 0.5 nmoles in 100 µL of water for injections mixed with ethanol (20% EtOH/80% water). For brain tumor imaging, U-87 MG spheroids of approximatively 500 µm diameter were implanted in the cortex of the right hemisphere and imaged 24 h after the administration of 0.5 nmoles of the compound **24** or **25**. In the same manner, major organs were harvested and subjected to small animal NIRF imaging (Fluorvivo, Indec Biosystem, Los Altos, CA, USA). A customized filter set (excitation, 510–550 nm; emission, 630–690 nm) was used for data acquisition. All fluorescence images were acquired with a 1-s exposure. The fluorescence intensity of each tissue was measured after subtraction of the background signal from an ROI of the same size and shape drawn over an area without any tissue using imageJ.

## 4. Conclusions

In this contribution, we developed a clickable *C*-glycosyl compound as central scaffold for the multistep synthesis of a cyanine-based dual PET/OI imaging probe. The radiofluorination was successfully performed by nucleophilic substitution of a mesylate leaving group, establishing the first direct radiofluorination of a cyanine-containing compound via the formation of a [^18^F]F-C bond. Two *c*(RGDfK) peptides were coupled to the two remaining positions of the glycosyl scaffold, and the resulting dimeric structures retained substantial affinity toward α_v_β_3_ with IC_50_ of 10 and 16 nM for compounds **23** and **24**, respectively. The original strategy reported in this paper permits the synthesis of a PET/OI dual probe which can be conjugated to any peptide of interest in the last step, allowing the versatility and imaging of a wide range of relevant biological targets. In vivo fluorescence imaging on U-87 MG engrafted nude mice displayed an orthotopic tumor accumulation with a fluorescence signal 40-fold higher in the tumor than in the healthy brain, as well as a high ectopic tumor uptake (ratio of 100 to 1 compared to the healthy brain). The ectopic tumor was resected, and confocal imaging of the tumor sections allowed the identification of tumor cells at high resolution. These preliminary results highlight the potential of compound **24** for glioblastoma cancer diagnosis using PET and image-guided surgery using OI. PET/OI bimodal imaging experiments will be reported in due course, opening a broad scope of clinical applications.

## Data Availability

Data is contained within the article and Appendix A.

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
