# Peer review of "Clickable C-Glycosyl Scaffold for the Development of a Dual Fluorescent and [18F]fluorinated Cyanine-Containing Probe and Preliminary In Vitro/Vivo Evaluation by Fluorescence Imaging"

_pharmaceuticals, 2022, doi:10.3390/ph15121490_

Round 1

Reviewer 1 Report

Topic is of relevance and study plan well thought out. 

Command of English language is mediocre, yet understandable since authors are clearly non-native. Suggest some proof-reading to improve grammatical and language errors

Author Response

Corrections are highlighted in yellow. 

Reviewer 2 Report

Well-organized manuscript, with indisputable scientific interest and interesting results. Only a few typos were detected by me and, regarding the structure, the manuscript does not match the journal's template. I just ask you to review this for publication.

Thus, without further comments, the manuscript, with regard to its content, is ready for publication.

Author Response

The manuscrit is now in the appropiate template.

Reviewer 3 Report

The authors investigated the synthesis of a cyanine-based dual PET/Optical imaging probe based on a versatile synthetic strategy and its direct radiofluorination via a F-C bond formation and also they evaluated in vitro and in vivo fluorescence imaging for the mice tumor. The manuscript is well prepared, the results and the experiment, and all figures are properly addressed. But, there are some issues in the manuscript which must be revised.

General Comments:

>>Introduction:

- What is the importance of this method (PET/OI) compared to PET/MRI or PET/CT? Please explain in the text. Can OI be used together with PET/CT? How much can it increase the accuracy of the work?

>>Results and discussion:

-The title of section 2 and 3 both have “discussion”. Because you have a discussion section, either the content related to the discussion should be separated from the results and moved to the next section, or the title of section 2 and 3 should be “results and discussion”, and the discussion section should not be separated.

- Is it possible to compare the results of your work with other imaging methods? especially for example, imaging with the PET/MRI or PET/CT?

- Page 6, Figure 2: How is the calculation error bar in the figure 2 obtained? Please, in figure 2, the symbols are enlarged and displayed in different shapes. Three symbols are circles.

- Page 6, Figure 3: What is the reason for normalizing to 10^6 cells? Does this number guarantee the accuracy of the results?

>>Conclusion:

- How practical can this method be for humans? Please more explain in this section.
